# Colicin E1 opens its hinge to plug TolC

S Jimmy Budiardjo[1], Jacqueline J Stevens[2], Anna L Calkins[3], Ayotunde P Ikujuni[2], Virangika K Wimalasena[2], Emre Firlar[4], David A Case[5], Julie S Biteen[3], Jason T Kaelber[4], Joanna SG Slusky[1,2]*

[1]Center for Computational Biology, The University of Kansas, Lawrence, United States; [2]Department of Molecular Biosciences, The University of Kansas, Lawrence, United States; [3]Department of Chemistry, University of Michigan, Ann Arbor, United States; [4]Rutgers CryoEM & Nanoimaging Facility and Institute for Quantitative Biomedicine, Rutgers University, Piscataway, United States; [5]Department of Chemistry and Chemical Biology, Rutgers University, Piscataway, United States

**Abstract** The double membrane architecture of Gram-negative bacteria forms a barrier that is impermeable to most extracellular threats. Bacteriocin proteins evolved to exploit the accessible, surface-exposed proteins embedded in the outer membrane to deliver cytotoxic cargo. Colicin E1 is a bacteriocin produced by, and lethal to, *Escherichia coli* that hijacks the outer membrane proteins (OMPs) TolC and BtuB to enter the cell. Here, we capture the colicin E1 translocation domain inside its membrane receptor, TolC, by high-resolution cryo-electron microscopy to obtain the first reported structure of a bacteriocin bound to TolC. Colicin E1 binds stably to TolC as an open hinge through the TolC pore—an architectural rearrangement from colicin E1's unbound conformation. This binding is stable in live *E. coli* cells as indicated by single-molecule fluorescence microscopy. Finally, colicin E1 fragments binding to TolC plug the channel, inhibiting its native efflux function as an antibiotic efflux pump, and heightening susceptibility to three antibiotic classes. In addition to demonstrating that these protein fragments are useful starting points for developing novel antibiotic potentiators, this method could be expanded to other colicins to inhibit other OMP functions.

## Editor's evaluation

Colicins are plasmid-encoded toxins produced by bacteria to kill closely related bacteria that compete for scarce resources. While the individual proteins used for binding and taking up colicins in target bacteria are well-known, the precise pathway that those toxins ultimately take to make it into the cell is unclear. Here the authors reveal the cryo EM structure of the outermembrane protein TolC in complex with colicin E1 toxin to shed new mechanistic insights into this toxin import mechanism.

*For correspondence:
slusky@ku.edu

Competing interest: The authors declare that no competing interests exist.

## Introduction

In Gram-negative bacteria, the outer membrane, cell wall, and cytoplasmic membrane form concentric barriers that protect the cell from extracellular threats. Of these protective structures, the outer membrane forms a particularly formidable obstacle (*Alexander and Rietschel, 2001*), owing to the impermeability of the lipopolysaccharide (LPS) layer that constitutes the outer membrane (*Kamio and Nikaido, 1976*). The primary means by which external molecules can gain access to the cell is through the ~100 variations of barrel-shaped proteins that are embedded in each bacterium outer membrane (*Freeman and Wimley, 2012*) and whose diverse functions include the transport of molecules across the membrane—specifically, the import of nutrients and metabolites and the export of toxins and waste.

**eLife digest** Bacteria are constantly warring with each other for space and resources. As a result, they have developed a range of molecular weapons to poison, damage or disable other cells. For instance, bacteriocins are proteins that can latch onto structures at the surface of enemy bacteria and push toxins through their outer membrane.

Bacteria are increasingly resistant to antibiotics, representing a growing concern for modern healthcare. One way that they are able to survive is by using 'efflux pumps' studded through their external membranes to expel harmful drugs before these can cause damage. Budiardjo et al. wanted to test whether bacteriocins could interfere with this defence mechanism by blocking efflux pumps.

Bacteriocins are usually formed of binding elements (which recognise specific target proteins) and of a 'killer tail' that can stab the cell. Experiments showed that the binding parts of a bacteriocin could effectively 'plug' efflux pumps in *Escherichia coli* bacteria: high-resolution molecular microscopy revealed how the bacteriocin fragment binds to the pump, while fluorescent markers showed that it attached to the surface of *E. coli* and stopped the efflux pumps from working. As a result, lower amounts of antibiotics were necessary to kill the bacteria when bacteriocins were present.

The work by Budiardjo et al. could lead to new ways to combat bacteria that will reduce the need for current antibiotics. In the future, bacteriocins could also be harnessed to target other proteins than efflux pumps, allowing scientists to manipulate a range of bacterial processes.

Because outer membrane proteins (OMPs) are accessible from outside the cell, bacteriophage and bacterial toxins have evolved to exploit OMPs to initiate delivering cargo across the outer membrane. Bacteriocins hijack the OMPs of a target bacterium to cross its impermeable outer membrane and kill the bacterium. Colicins are *Escherichia coli*-specific bacteriocins, protein toxin systems through which bacteria engage in bacterial warfare with other, similar bacteria. Colicins can be divided into groups A and B based on the requirement for which periplasmic protein systems they rely on to enter cells. The cytotoxicity of group A colicins requires the Tol system (*Nagel de Zwaig and Luria, 1967*) which is coupled to the proton motive force for its role in maintaining membrane integrity (*Lazzaroni et al., 1989*). Colicins differ in their receptor targets and killing mechanisms. Most colicins share a common tri-domain architecture, comprising the following components: (i) an N-terminal translocation (T) domain, (ii) a receptor-binding (R) domain, and (iii) a C-terminal cytotoxic (C) domain (*Figure 1A*). Much of what is known of group A colicin import has been determined through studies of the colicins E3 and E9 as reviewed by *Cramer et al., 2018*. For colicins E3 and E9, import is initialized by high-affinity R domain binding to the vitamin B12 transporter, BtuB (*Kurisu et al., 2003*; *Housden et al., 2005*; *Francis et al., 2021*); this binding localizes the colicin onto the outer membrane. Once the colicin is tethered to the outer membrane surface, the T domain of ColE3 and ColE9 initiates translocation using the secondary OMP receptor/translocator OmpF to access the Tol-Pal system in the periplasm. For most colicins, the T domain requires an OMP distinct from the R domain target for colicin translocation (*Cascales et al., 2007*). A handful of colicins have been structurally characterized with their OMP counterparts (*Kurisu et al., 2003*; *Sharma et al., 2007*; *Buchanan et al., 2007*). Colicin E9 bound to both BtuB and OmpF was structurally resolved as a complete translocon complex (*Francis et al., 2021*). The structure of the full ColE9 translocon showed that the T domain can fully penetrate deeply into lumen of the outer membrane receptors while the R domain interacts with the extracellular loop regions of its receptors. Studies of the bacteriocin pyocin S5 from *Pseudomonas aeruginosa* suggest that bacteriocin architectures and mechanisms may be conserved across all Gram-negative species (*Behrens et al., 2020*).

Colicin E1 is a bacteriocin produced by *E. coli* that enters the periplasm of neighboring cells and forms a pore on the cytoplasmic membrane leading to membrane depolarization and cell death. Unlike any of the other E colicins, the T domain of Colicin E1 binds to TolC (*Benedetti et al., 1991*), the outer membrane component of the acridine efflux pump, and the R domain binds BtuB (*Benedetti et al., 1989*), the primary receptor shared by all E colicins. A high-resolution X-ray structure exists of only the cytotoxic domain of colicin E1 (*Elkins et al., 1997*) but not the T or R domains, thus no mechanistic insights can be gained about initial steps of import. Domain swapping experiments (*Benedetti et al., 1991*) between colicin A and E1 identified regions of the protein that interact with

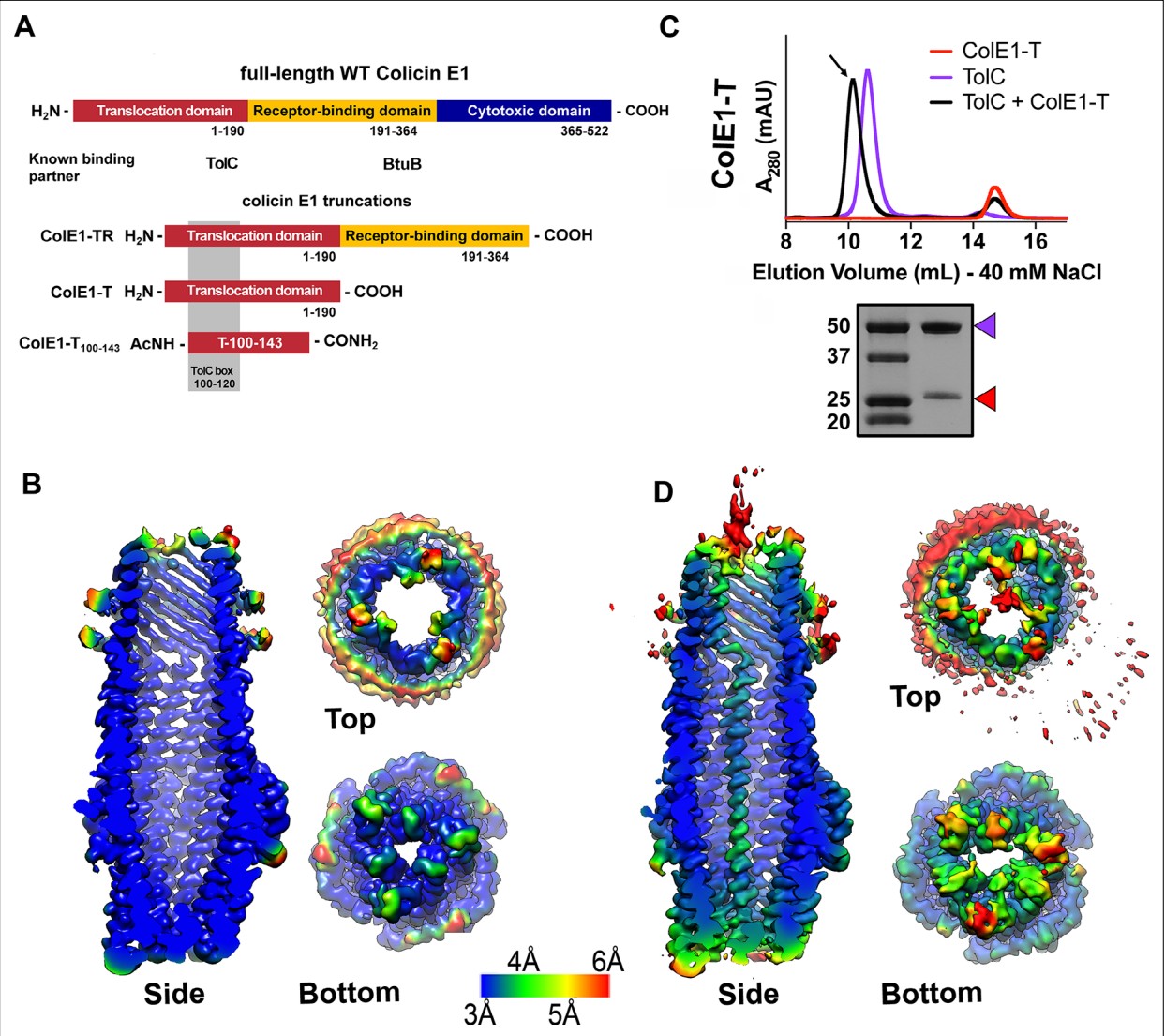

**Figure 1.** Colicin E1-T binds within the lumen of TolC. (**A**) Architecture of full-length colicin E1 showing domains and their known binding partners. Three truncation constructs used in this study. (**B**) CryoEM structure of TolC embedded in nanodiscs. Side, top, and bottom views are colored by local resolution, as computed in cryoSPARC from the final half-maps. The side view is cropped to display the particle interior. (**C**) SEC chromatogram of ColE1-T (red line) and TolC (purple line). The arrow indicates the co-elution (black line) fractions (eluted in buffer containing 40 mM NaCl) that were analyzed by SDS-PAGE. On the SDS-PAGE gel (bottom), red arrows indicate the presence of the colicin E1 construct that has co-eluted with TolC. SEC chromatogram of TolC purification (*Figure 1—figure supplement 1*) and co-elution with 200 mM NaCl (*Figure 1—figure supplement 2*). (**D**) The CryoEM structure of ColE1-T bound to TolC and colored by local resolution as in (**B**). CryoEM data collection, refinement, and validation statistics in *Figure 1—source data 1*. cryoEM, cryo-electron microscopy; SEC, size exclusion chromatography.

The online version of this article includes the following source data and figure supplement(s) for figure 1:

**Source data 1.** CryoEM data collection, refinement, and validation statistics.

**Figure supplement 1.** Sample SEC chromatogram of TolC purification after refolding.

**Figure supplement 2.** Co-elution of ColE1-T with TolC at 200 mM NaCl (black).

specific OMPs. Early studies of pore-forming colicins, including E1, showed that cytotoxic-domain-induced K$^+$ efflux on the inner membrane can be reversed after it has begun by subsequent addition of trypsin to cells. This, in addition to trypsin's inability to cross the outer membrane, led to the belief that portions of colicin remain tethered to their OMP partners as the cytotoxic domain depolarizes the cytoplasmic membrane.(*Plate and Luria, 1972*; *Dankert et al., 1980*; *Dankert et al., 1982*; *Bénédetti et al., 1992*).

In order to understand the early stages of colicin E1 import, functional studies of truncations of colicin E1 which lack the cytotoxic domain have been characterized in vitro. Residues 100–120 of colicin E1 (termed the 'TolC box,' *Figure 1A*) have been shown to be necessary but not sufficient for binding TolC. Peptides that include the TolC box co-elute with TolC (*Zakharov et al., 2016*) and disrupt channel conductance in planar lipid bilayer membranes (*Zakharov et al., 2004*; *Zakharov et al., 2016*). Moreover, *E. coli* exposure to TolC-box-containing peptides can prevent subsequent binding and cytotoxicity of full-length colicin E1 (*Jakes, 2017*). CD of the T domain indicates that it exists as a helical hairpin (closed hinge) in solution similar to other colicin T domains (*Wiener et al., 1997*; *Housden et al., 2021*) and that the proline at the center of the TolC box forms its apex (*Zakharov et al., 2016*; *Housden et al., 2021*). This measurement led to the proposal, known as the 'pillar model,' (*Zakharov et al., 2016*) that the T domain inserts into the TolC barrel as a helical hairpin where the N and C-termini are pointing to the cell exterior. According to this model, the hairpin stuck in TolC acts as a buttress to facilitate the C terminal-first cytotoxic domain entry directly through the membrane where it is possibly cleaved from the TR domain to reach its inner membrane target similar to ColE7 (*Shi et al., 2005*) and ColD (*de Zamaroczy et al., 2001*).

A competing model for colicin E1 import, known as the 'total thread' model (*Cramer et al., 2018*; *Zakharov et al., 2016*; *Housden et al., 2013*; *Francis et al., 2021*) posits that the protein unfolds and passes through TolC N-terminus-first as an unstructured peptide and binds to TolA (*Pilsl and Braun, 1995*) in the periplasm. In this model, the binding between the intrinsically unstructured colicin N-termini and periplasmic proteins (*Jakes, 2017*; *Housden et al., 2013*) creates a pulling force that results in the translocation of the whole colicin.

Here, we use cryo-electron microscopy (cryoEM) to solve the high-resolution structure of a bacteriocin, colicin E1, bound to TolC. We find that colicin E1 binds stably to TolC, not as a helical hairpin but as a single-pass folded helix with the N-terminus inside the periplasm. Additionally, we find that the ColE1 TR domain binds TolC in vivo as well. Using single-molecule fluorescence microscopy, we find that ColE1-TR, lacking the cytotoxic domain, remains stalled on the outer membrane and does not fully translocate into cells. Lastly, we leverage this stalling of ColE1 to block the native TolC function as an antibiotic efflux pump. Because they are accessible from outside the cell, OMPs are attractive targets for the development of novel antibiotics, and research has begun to reveal the therapeutic potential of interfering with OMP structure and function. (*Storek et al., 2018*; *Kaur et al., 2021*).

Here by determining the structure and mechanism of ColE1 insertion, we establish an alternative approach for targeting OMPs—the development of molecular plugs that block OMP pores. Such plugs would allow for the manipulation of bacterial transport, providing a means of either starving the bacterium by preventing the import of valuable nutrients or poisoning the bacterium by preventing the export of toxins. Through real-time efflux assays and minimum inhibitory concentration (MIC) experiments, we find that ColE1-TR and ColE1-T are able to inhibit TolC-mediated efflux. We find that this plugging of TolC reduces the amount of antibiotics required to inhibit the bacterial growth—indicating that this colicin E1 fragment reduces the antibiotic resistance conferred by TolC.

## Results

### CryoEM structure of TolC embedded in nanodiscs

To determine the structural details of colicin E1 binding to TolC, we solved the cryoEM structure of TolC embedded in nanodiscs with and without added ColE1-T (PDB 6WXH and 6WXI, respectively). We recently reported a high-yield TolC preparation by refolding from inclusion bodies (*Budiardjo et al., 2021*) and inserted refolded TolC into nanodiscs. The cryoEM structure of refolded TolC alone (*Figure 1B*) is similar to the previously published crystal structures of natively derived TolC alone (*Koronakis et al., 2000*) or in complex with hexamminecobalt (*Higgins et al., 2004*) but with more splayed loops at the periplasmic opening (residues 165–175). There was local variation in resolution within the model with lower resolution associated with the extracellular/periplasmic ends and the nanodisc scaffold protein (*Figure 1B*). The lower residue resolution at these apertures may indicate dynamics not captured in the X-ray crystal structures.

We next determined the binding of colicin E1-T domain to capture the complex for structure determination. Residues 1–190 (ColE1-T) which span the translocation domain are known to bind to TolC (*Figure 1A*). We identified ColE1-T:TolC binding in vitro via co-elution by size exclusion

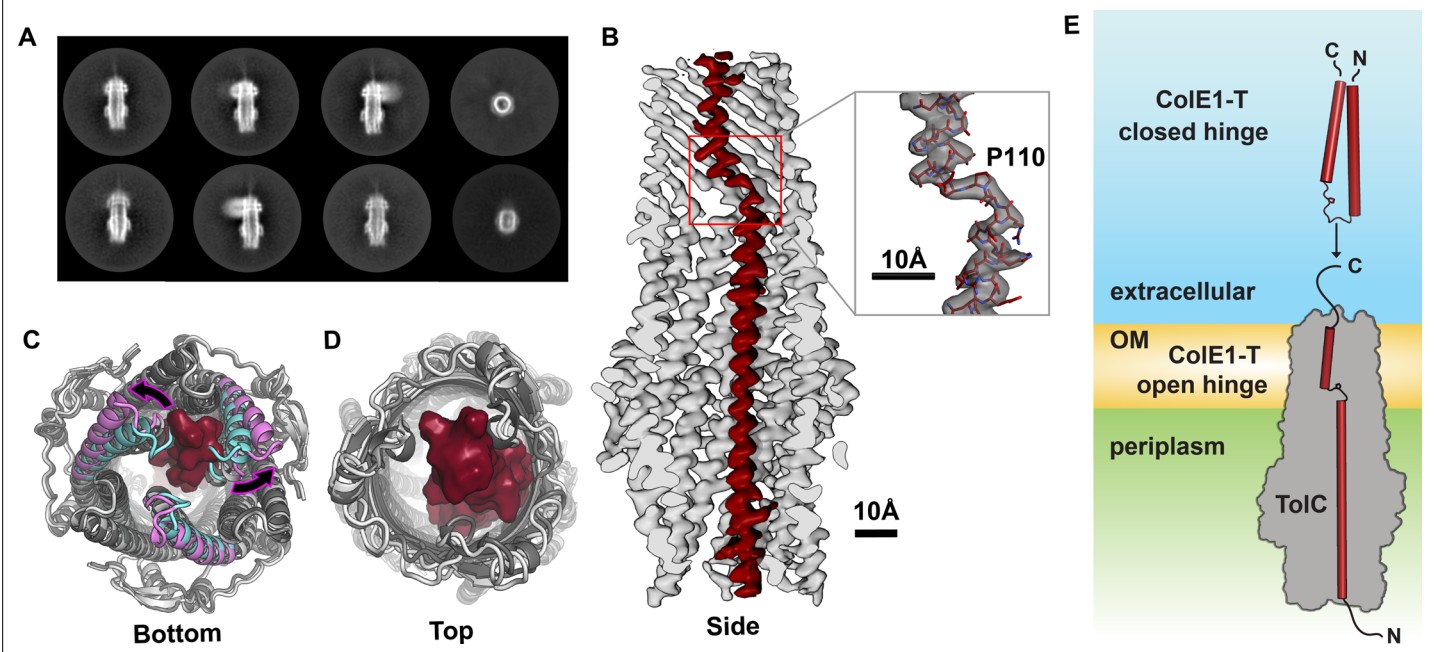

**Figure 2.** Colicin E1 binds to TolC as a single-pass kinked helix. (**A**) 2D class averages of ColE1-T bound to TolC embedded in nanodiscs. (**B**) Cutaway side view of the TolC (light gray) interior. ColE1-T (red) binds asymmetrically in an open-hinge conformation. The hinge region of ColE1-T including residue P110 shown in stick representation (inset). (**C, D**) The apo cryoEM structure of TolC (light gray) compared to holo cryoEM structure (dark gray). (**C**) Periplasmic aperture. Two of the TolC helices in the bound structure (magenta) dilate outwards compared to those two helices in the unbound state (cyan). (**D**) The extracellular aperture remains more similar between the bound and unbound structures. (**E**) In the unbound state, ColE1-T (red cylinders) exists as a closed hinge. When bound, ColE1-T is in an open-hinge conformation through TolC with the N-terminus into the pore of TolC (gray). The side chain of the hinge proline is shown making a small pentagon in both the closed and open hinge states. The model of ColE1-T in its solution state was built using AlphaFold (*Jumper et al., 2021*) and is consistent with previous experiments (*Zakharov et al., 2016*). cryoEM, cryo-electron microscopy.

chromatography (SEC) as previously described (*Zakharov et al., 2016*). When ColE1-T and TolC were mixed, we observed a leftward shift in the TolC peak and a decrease in intensity associated with the ColE1-T peak indicating that a subset of the population had migrated with TolC (*Figure 1C*). We analyzed the peak (*Figure 1C* arrow) using SDS-PAGE and found the presence of both ColE1-T and TolC. The unimodal shifted peak observed with ColE1-T:TolC indicates that there is a single species of fully bound TolC. When resolved by cryoEM, addition of ColE1-T to TolC breaks the threefold channel symmetry and the additional protein is observed running all the way through the TolC barrel as a single-pass, all-α-helical chain spanning more than 130 Å (*Figure 1D*). The maps refined to nominal global resolutions of 2.84 Å and 3.09 Å for the TolC and ColE1-T:TolC, respectively (*Figure 1—source data 1*).

The ColE1 T domain inserts into TolC with its amino terminus pointing inwards through the periplasmic iris, and 2D class averages show that the carboxy terminus of the helix continues and projects out into the extracellular space (*Figure 2A*). About 85 of the 190 ColE1-T residues could be modeled to this density (residues 46–131) (*Figure 2B*). No such regular protrusion was seen for the glycine-rich ColE1 amino-terminus, which we expect to be disordered in the periplasmic space (*Housden et al., 2013*; *Housden et al., 2018*; *Zakharov et al., 2016*). The ColE1 chain binds TolC asymmetrically, primarily contacting only one of the three TolC chains.

Compared to the unbound state, asymmetric ColE1 binding dilates the periplasmic TolC aperture (*Figure 2C*) and, makes only minor adjustments to the loops on the extracellular aperture (*Figure 2D*) so that they can accommodate ColE1 in the absence of any other proteins or motive force.

In solution, unbound colicin E1 is a two-helix hairpin as indicated by far UV circular dichroism (CD) (*Zakharov et al., 2016*). A structural model built using AlphaFold (*Jumper et al., 2021*) also predicts the closed hinge conformation with proline 110 at the apex (*Figure 2E*, top). However, our cryoEM reconstruction of colicin E1 in complex with TolC shows that ColE1-T hinges open at the TolC box to an extended conformation upon binding to TolC (*Figure 2B and E*, bottom). The ColE1-T helices have

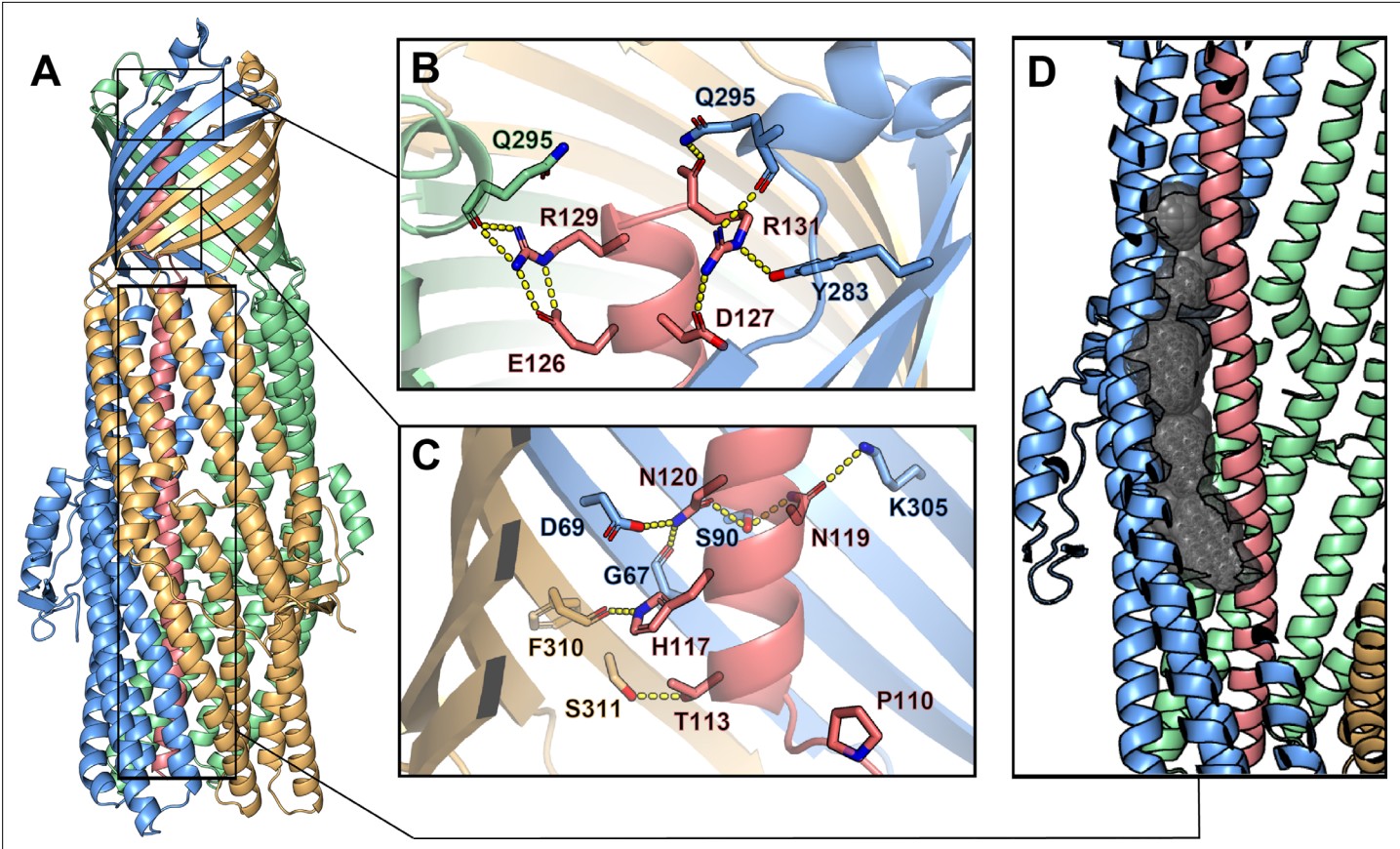

**Figure 3.** Inter-chain contacts between ColE1 T and TolC. (**A**) Molecular dynamics simulation refined structure of the ColE1-T:TolC complex colored by chain (ColE1-T: light red, TolC: light blue, light green, light gold). (**B, C**) Polar interaction network of ColE1-T and TolC on the C-terminal side of the proline kink in the β-barrel region. Locations of previous mutagenesis data near the TolC box (*Figure 3—figure supplement 1* and *Figure 3—video 1*). (**B**) Near the extracellular opening of TolC and (**C**) in the TolC barrel near the transition to the TolC periplasmic helical region. Residues involved in forming polar interactions and proline 110 are shown in sticks. These interactions support binding such that even when the hinge proline is mutated to alanine (P110A) binding is maintained (*Figure 3—figure supplement 2*). (**D**) ColE1-T spanning the TolC helical barrel does not make full contact with the side of the TolC barrel. The cavity (black spheres) between colicin (light red) and one chain of TolC (light blue) was detected using GHECOM (*Kawabata, 2010*; *Kawabata, 2019*; *Kawabata and Go, 2007*).

The online version of this article includes the following video and figure supplement(s) for figure 3:

**Figure supplement 1.** Cut away side-view of TolC:ColE1-T complex.

**Figure supplement 2.** ColE1-TP110A retains binding to TolC.

**Figure 3—video 1.** Pan through the atomic model of the TolC/colE1-T complex, from the extracellular aperture to the periplasmic aperture.
https://elifesciences.org/articles/73297/figures#fig3video1

---

a short kink in the middle around proline 110, precisely at the proposed turn location in the hairpin model (*Figure 2B*, inset). This area is also the center of the TolC box (*Figure 1A*) that has been known to be critical for the ColE1:TolC interaction (*Jakes, 2017*; *Zakharov et al., 2016*).

## Colicin E1 makes residue – specific interactions with TolC

TolC forms a large rigid conduit in the outer membrane and periplasm (*Figure 3A*) with a water-filled lumen. The hydrophilic nature of the channel (*Dhar et al., 2021*) coupled with the inflexibility of the outer-membrane-embedded barrel largely precludes the formation of any hydrophobic interfaces that are typically the basis of protein-protein interfaces involving helical peptides (*Eisenberg et al., 1982*). Yet, we did observe specific polar contacts between colicin E1's TolC box and the TolC barrel.

To obtain the most accurate atomic model for interpretation of atomic interactions between peptides, we utilized map-restrained molecular dynamics in model refinement (*Wu et al., 2013*). The refined model had improved chemical plausibility (for instance, the MolProbity [*Williams et al., 2018*]

score improved from 1.25 to 0.50) and polar contacts were more easily identified. Because this refinement improved the concordance between the map and model, we conclude that the use of cryoEM density as a restraint was successful in preventing overfitting. Specifically, the EMRinger (*Barad et al., 2015*) score improved from 2.77 to 3.81 and the map-model FSC=0.5 improved from 3.44 Å to 3.37 Å.

The atomic model showed significant polar interactions form between colicin E1 and TolC at the apertures and around the TolC box. The acidic patch of colicin E1 near the extracellular aperture contains two arginine residues (R129 and R131) that form hydrogen bond networks with one TolC chain each (*Figure 3B*). In addition, colicin E1 residues T113, A116, N119, and N120 (all part of the TolC box) establish polar interactions with TolC residues G67, D69, S90, and K305 on one TolC chain (*Figure 3C*, light red and light blue) while colicin H117 interacts with TolC F310 on a neighboring TolC chain (*Figure 3C*, light red and light gold). These contacts are strong enough that even when the proline hinge of the TolC-box is mutated (P110A) ColE1-T still binds TolC (*Figure 3—figure supplement 2*).

By contrast with the β-barrel of TolC, in the interior of the periplasmic helical barrel of TolC, colicin E1 does not make full contact with the α-helical barrel interior and there is a gap between the two proteins (*Figure 3D*, *Figure 3—video 1*). This is in agreement with previous studies that showed a colicin truncation (residues 1–100), that ends just before the beginning of the TolC box, does not bind to TolC (*Zakharov et al., 2016*) or prevent cytotoxicity of full-length colicin in cells (*Jakes, 2017*) as this region of the colicin does not make strong interactions with TolC (*Figure 3D*).

Moreover, this structure is consistent with reports that mutations at TolC site G65 or S279 abrogate ColE1 activity (*Masi et al., 2007*) as these residues are contact sites between TolC and ColE1 (*Figure 3—figure supplement 1*, and *Figure 3—video 1*). Conversely, the absence of any effect of mutating colicin R103 and R108 (*Jakes, 2017*), is consistent with interaction those residues have with the solvent in the barrel rather than with TolC. The newly solved structure, in combination with previous work (*Jakes, 2017*; *Zakharov et al., 2016*), indicates the specificity of colicin E1 binding to TolC is encoded in the portion that binds to TolC β-barrel.

## Membrane localization of ColE1 truncations

Since we find the ColE1 T and TR domains bind to TolC in vitro, we investigated if colicin E1 fragments bind to the native TolC in *E. coli* cells. Though ColE1 C domain activity—depolarizing the inner membrane—requires the C domain to pass through the outer membrane (*Brunden et al., 1984*), it remains unclear if the ColE1-TR domains translocate as well. We first determined that TolC binds to the TR domain in vitro similarly to TolC binding to the T domain. ColE1-TR includes residues 1–364 (ColE1-TR) contains the N-terminal portion that binds to TolC and the C-terminal portion that binds to BtuB. ColE1-TR aggregates at the salt concentration (40 mM) used for the co-elution experiment used for ColE1-T so we increased the NaCl concentration to 200 mM. ColE1-T still binds to TolC at 200 mM NaCl although to a much lesser degree (*Figure 1—figure supplement 1*). Unlike ColE1-T (*Figure 1C*), ColE1-TR shows a bimodal distribution indicative of a mixed population of bound and unbound TolC (*Figure 4A*).

We assessed whether ColE1-T and ColE1-TR translocate into the cell or localize on the cell surface using an extracellular protease digestion assay (*Besingi and Clark, 2015*). Colicin fragment stalling on the outer membrane surface would render the colicin fragments susceptible to digestion when trypsin is in the extracellular environment. Conversely, if the colicin fragment translocates into the intracellular environment, the outer membrane will shield the protein from digestion by trypsin.

Colicin fragment localization was probed through western blot analysis of specific colicin fragments with C-terminal polyhistidine-tags. Cells were incubated with these colicin fragments and subsequently exposed to two different concentrations of trypsin. If the protein enters the cell, the outer membrane shields it from trypsin digestion, and a corresponding band is present on the western blot. Periplasmic protein SurA was used as a membrane integrity control. SurA digestion indicates that the periplasm was accessible to trypsin and a SurA band indicates that the periplasm was shielded from trypsin. As a trypsin activity control, half of the cell sample was lysed (lysed cell condition) exposing all cellular compartments to trypsin.

When probing for the C-terminal epitope by immunoblotting, there was no detectible ColE1-T binding (*Figure 4B*, left), though there was detectible binding of ColE1-TR (*Figure 4B*, right). Moreover, we found that if ColE1-TR is incubated with cells and subsequently treated with two different

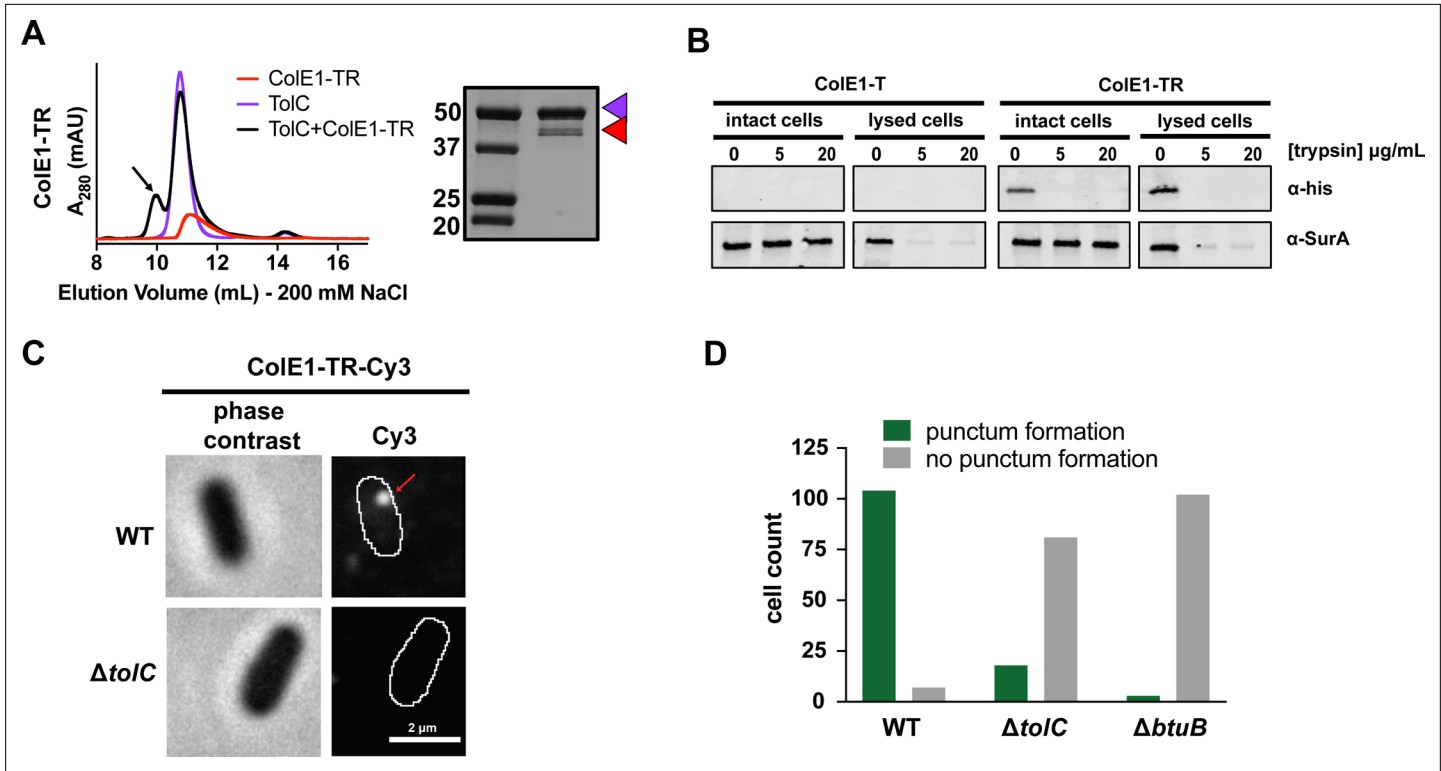

**Figure 4.** Colicin E1-TR localizes on the outside of the cell. (**A**) SEC chromatogram of ColE1-TR (red line) and TolC (purple line). The arrow indicates the co-elution (black line) fractions that were analyzed by SDS-PAGE. On the SDS-PAGE gels (right), red arrow indicates the presence of colicin E1 construct that has co-eluted with TolC (purple arrow). (**B**) Extracellular protease digestion assay with two colicin E1 truncation constructs incubated with *Escherichia coli*, each construct is labeled with a C-terminal His-Tag. Periplasmic SurA was used as a membrane integrity control. After incubation with colicin fragments, cells were left intact or lysed with lysozyme. Proteins degraded by trypsin in intact cells are not protected by the outer membrane. (**C**) Fluorescence image (right) of ColE1-TR-Cy3 overlaid on outlines of living *E. coli* cells from phase-contrast microscopy (left) for WT and Δ*tolC*. Red arrow points to a punctum. A larger variety of images is also available (*Figure 4—figure supplement 1*). Similar localization was seen for ColE1-TR-GFP (*Figure 4—figure supplement 2*) demonstrating that the effect was not caused by the fluorophore, and ColE1-TR$_{\Delta1-40}$-Cy3 (*Figure 4—source data 1*, *Figure 4—figure supplement 3*) demonstrating that the effect is not TolA dependent. The puncta were stable and were found to remain intact for more than 5 min. The punctum shown here remained intact for at least 30 s (*Figure 4—video 1*). (**D**) Cell counts where ColE1-TR-Cy3 punctum formation was observed for WT, Δ*tolC*, and Δ*btuB*. Number of cells observed, n=111, 91, 105, respectively. SEC, size exclusion chromatography.

The online version of this article includes the following video, source data, and figure supplement(s) for figure 4:

**Source data 1.** Comparison of percent cells with puncta and average number of puncta per cell between ColE1-TR-Cy3 and ColE1-TR$_{\Delta1-40}$-Cy3 with WT Δ*tolC* and Δ*btuB*.

**Figure supplement 1.** Additional examples of cells observed during imaging described in *Figure 4C and D*.

**Figure supplement 2.** Single-molecule microscopy.

**Figure supplement 3.** Examples of cells observed with ColE1-TR$_{\Delta1-40}$-Cy3.

**Figure 4—video 1.** A representative 30-s movie of ColE1-TR bound to a live cell.

https://elifesciences.org/articles/73297/figures#fig4video1

trypsin concentrations, the ColE1-TR band disappeared, indicating that the colicin E1 fragment was localized to the outer membrane surface (*Figure 4B*, right). The periplasmic control SurA was not degraded at any trypsin concentration unless the cells were lysed before trypsin digestion (*Besingi and Clark, 2015*).

We further probed the interaction between the cell membrane and surface-localized ColE1-TR with single-molecule fluorescence microscopy using the fluorescent dye Cy3 (*Al-Husini et al., 2020*; *Tuson and Biteen, 2015*). When ColE1-TR-Cy3 was added to the extracellular environment of WT BW25113 *E. coli* (containing TolC), distinct puncta (*Figure 4C*, right) formed on 94% of the cells (number of cells, n=111) (*Figure 4D*); cells with observed puncta most often featured a single punctum, though a small fraction had two puncta. On average, the WT cells had 1.2 puncta. In a Δ*tolC* strain, puncta were

observed on only 18% of cells (n=99) (*Figure 4C*, right; *Figure 4D*); on average, the Δ*tolC* cells had 0.2 puncta. As a Δ*btuB* control strain, we used BL21 (DE3) which is known to have a premature stop codon at BtuB residue 58 (*Studier et al., 2009*). Puncta were observed on only 3% of cells lacking BtuB (n=105) (*Figure 4D*, *Figure 4—figure supplement 2A* ); on average, the Δ*btuB* cells had 0 puncta.

The bright ColE1-TR-Cy3 puncta photobleach over time as they are observed in the fluorescence microscope, ultimately leaving a single-molecule fluorescing before complete photobleaching. We therefore divided the brightness of each punctum by the brightness of that single molecule to estimate the number of molecules in each cluster (*Bakshi et al., 2012*). In WT and Δ*tolC* cells that featured puncta, ColE1-TR form puncta consistent with ~20 molecules; we never detect an isolated single ColE1-TR molecule on either WT or Δ*tolC* cells before photobleaching. The observed size and number of molecules agree with previous studies of BtuB clusters (*Chavent et al., 2018*; *Rassam et al., 2015*) and the punctum locations within the cell were variable. To rule out punctum formation as an artifact of Cy3 conjugation, we found ColE1-TR-GFP displayed the same cluster formation as ColE1-TR-Cy3 (*Figure 4—figure supplement 2B*).

Because we usually only see one punctum per cell, we anticipate that some BtuB and TolC may remain unbound because of the geometric constraints of punctum formation. ColE1 must engage both TolC and BtuB simultaneously. Therefore, both receptors must be in close proximity, similar to the ~50 Å proximity of BtuB to OmpF when both bound to another type A colicin, ColE9 (*Francis et al., 2021*). Given the two-dimensional surface of a membrane, there are restrictions to how many proteins can be within 50 Å of any other protein. For the clusters we see to occur, when BtuB clusters together in groups of approximately 20 (*Figure 4*) TolC must also cluster in groups of approximately 20, ultimately requiring approximately 40 proteins in a relatively small area. The competition between the clustering of BtuB and the requirement for proximity of BtuB and TolC for binding could lower the number of BtuB/TolC sites available for the T and R domains of colicin E1 due to the relative TolC and BtuB geometries needed for ColE1 binding.

Fluorescently labeled pyocins, the colicin analog in *Pseudomonas*, have previously been used to detect translocation across the outer membrane of *P. aeruginosa* (*White et al., 2017*). Here, we use an analogous experiment with fluorescently labeled ColE1 to determine cellular localization in *E. coli*. In time courses, bound ColE1-TR puncta remained immobile for >5 min (*Figure 4—video 1* corresponds to 30 s of data used to attain the WT image in *Figure 4C*). This is consistent with the continued association of ColE1-TR with membrane-embedded BtuB, which has limited mobility (*Kleanthous et al., 2015*). This result indicates that ColE1-TR does not fully translocate (*White et al., 2017*), because if ColE1-TR entered the periplasmic space it would freely diffuse on these timescales.

Colicin constructs lacking the R domain (ColE1-T-Cy3) showed no detectable binding either to WT or Δ*tolC* cells (*Figure 4—figure supplement 2C*), indicating that the TolC-ColE1-T interaction is weaker than the BtuB-ColE1-TR interaction, likely due to the absence of the R domain-BtuB interaction.

TolA is known to be required for ColE1 cytotoxicity (*Nagel de Zwaig and Luria, 1967*; *Pilsl and Braun, 1995*). ColE1 lacking the TolA box at the N-terminal T-domain would likely be translocation incompetent due to uncoupling from TolA. Similarly, there is a loss of biological activity due to mutations in ColE9 N-terminal region that indirectly interacts with TolA (*Loftus et al., 2006*; *Garinot-Schneider et al., 1997*). Biological inactivation has also been caused by mutation to the Tol-Pal interacting regions of ColE3 (*Escuyer and Mock, 1987*) and ColA (*Pilsl and Braun, 1995*). We found that ColE1-TR$_{\Delta 1-40}$ binds to the cell surface of WT, Δ*tolC*, and Δ*btuB* in a similar manner as ColE1-TR (*Figure 4—source data 1*, *Figure 4—figure supplement 3*). Cell surface localization of ColE1-TR matching that of translocation deficient ColE1-TR$_{\Delta 1-40}$ suggests that ColE1-TR puncta are not within the cell and likely stalled on the outer membrane receptor. The requirement for TolA may serve an alternate role in translocation as we consider in the Discussion section.

## ColE1-TR inhibits efflux and makes *E. coli* more susceptible to antibiotics

Since the colicin truncations are stalled on their respective OMP receptors, we investigated if this stalling could disrupt native TolC efflux. Real-time efflux inhibition by colicin E1 fragments was assessed using a live-cell assay with N-(2-naphthyl)-1-naphthylamine (NNN)-dye, which passively diffuses into cells and is effluxed by the AcrAB-TolC efflux pump. NNN fluoresces when it is localized in the hydrophobic interior of one of the membranes (*Bohnert et al., 2011*) thereby acting as a reporter for the

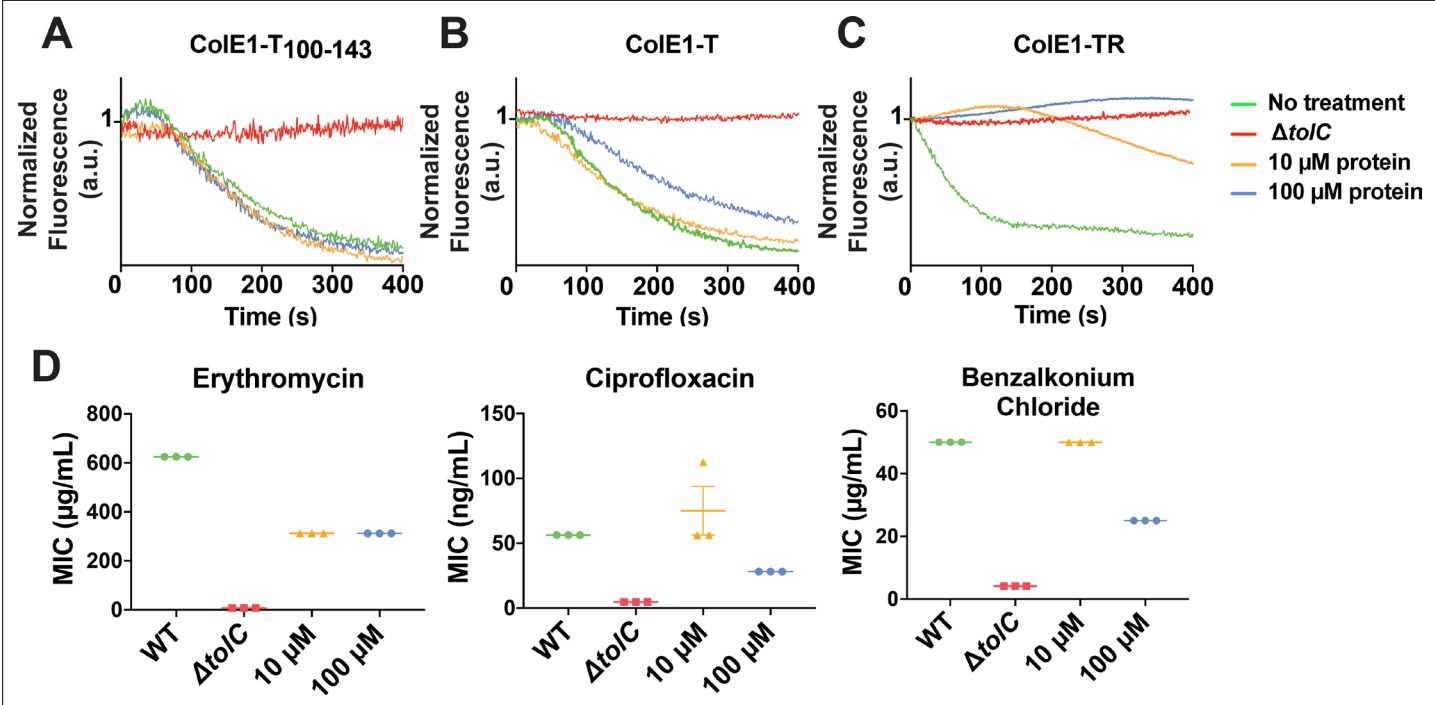

**Figure 5.** Colicin E1 fragments inhibit efflux and potentiate antibiotics. (**A–C**) Effect of colicin E1 fragments on efflux: WT with no protein (green), ΔtolC (red), WT + 10 µM (orange), and WT + 100 µM protein (blue). (**A**) ColE1-T$_{100-143}$, (**B**) ColE1-T, and (**C**) ColE1-TR. All efflux experiments used 10 µM of fluorescent reporter dye NNN. (**D**) Antibiotic susceptibilities in the absence (green) and presence of ColE1-TR added to WT at 10 µM (orange) and 100 µM (blue). MICs for ΔtolC (red) are included as a reference. Data shown from three biological replicates reported as the mean and individual data points. Bars indicate standard error of the mean. Data values shown in *Figure 5—source data 1*. MIC, minimum inhibitory concentration.

The online version of this article includes the following source data and figure supplement(s) for figure 5:

**Source data 1.** Mean minimum inhibitory concentrations (MICs) of antimicrobials in the presence of ColE1-TR.

**Figure supplement 1.** Minimum inhibitory concentration (MIC) of erythromycin with ColE1-TR TolA box deletion construct.

relative rates of diffusion and efflux. NNN efflux can be turned off by the protonophore CCCP, which neutralizes the proton motive force and allows NNN to accumulate within the cell. Active efflux can then be monitored by the fluorescence decay once the proton motive force is reenergized by glucose addition (*Bohnert et al., 2010*; *Seeger et al., 2008*; *Bohnert et al., 2011*; *Iyer et al., 2015*; *Misra et al., 2015*).

We assessed the ability of colicin E1 truncations to plug TolC by monitoring real-time NNN efflux. Cells exposed to ColE1-T$_{100-143}$ did not show lower real-time efflux than untreated cells (*Figure 5A*). This observation is notable in light of the fact that, in previous conductance studies, similar peptides were shown to bind TolC (*Jakes, 2017*; *Zakharov et al., 2016*) and to occlude the channel (*Zakharov et al., 2004*; *Zakharov et al., 2016*).

Cells exposed to high concentrations of ColE1-T showed less decay in final fluorescence than untreated cells, indicating that ColE1-T partially inhibits efflux (*Figure 5B*). Finally, exposure to high concentrations of ColE1-TR produced no decrease in fluorescence, showing full inhibition of efflux (*Figure 5C*).

Because ColE1-TR completely inhibits NNN efflux, we evaluated the capacity of ColE1-TR to potentiate antibiotics representing three different classes that are known TolC substrates: ciprofloxacin, erythromycin, and benzalkonium chloride (from fluoroquinolones, macrolides, and quaternary ammonium compounds, respectively). An effective TolC plug will reduce the concentration required to inhibit growth as antibiotics remain trapped within the cell. WT *E. coli* cells that were exposed to 100 µM ColE1-TR with each of these antibiotics showed significantly lower MICs, the lowest concentration of antibiotic that inhibits visible growth, than cells exposed to the antibiotics alone (*Figure 5D*): exposure to 100 µM ColE1-TR made WT *E. coli* ~2–7-fold more susceptible to these antibiotics (*Figure 5—source data 1*). Exposure of WT cells to 10 µM ColE1-TR was sufficient to potentiate erythromycin.

TolA is normally involved in outer membrane invagination during cell division (*Gerding et al., 2007*). Like Colicin E1, Colicin E9 contains an intrinsically unstructured N-terminus that engages the Tol-Pal system once it penetrates into the periplasm (*Loftus et al., 2006*). Engagement of the Tol-Pal system by the N-terminal region of colicin E9 disrupts native Tol-Pal system function leading to outer membrane defects (*Rassam et al., 2018*). Outer membrane defects may lead to enhanced antibiotic susceptibility. To rule out outer membrane-mediated defects as the main contributor to the reduction in MICs seen in *Figure 5*, we tested the effect of using a truncation of ColE1-TR (ColE1-TR$_{\Delta1-40}$) lacking the N-terminal 40 residues including the TolA box. The MIC of ColE1-TR and ColE1-TR$_{\Delta1-40}$ is identical (*Figure 5—figure supplement 1*) indicating that ColE1-TR engagement of TolA does not contribute to the observed antibiotic susceptibility.

## Discussion

There are two conflicting models of ColE1 translocation: (1) the total thread model in which the entire colicin unfolds and threads through TolC (*Cramer et al., 2018*; *Jakes, 2017*; *Francis et al., 2021*) and (2) the pillar model, in which colicin E1 inserts into TolC as a helical hairpin, facilitating LPS-mediated, self-translocation of the colicin cytotoxic domain (*Zakharov et al., 2016*). Our data support aspects of both models. The belief in a closed hinge conformation of bound ColE1 prompted two arguments against the total thread mechanism: (1) the ColE1-T closed hinge conformation is too wide to fit through TolC and (2) both ends would face the extracellular milieu (*Zakharov et al., 2016*; *Cramer et al., 2018*). The ColE1-T hinge opening in the bound state—which was unanticipated by the pillar model—resolves these objections: colicin threads into the TolC barrel with the amino terminus pointing into the periplasm, as the total thread model hypothesized. Our trypsin digests show ColE1-TR stalls at the outer membrane of *E. coli* cells and the single-molecule fluorescence images confirm that ColE1-TR remains clustered and does not diffuse as would be expected if entering the periplasm or cytoplasm. These puncta form regardless of the presence or absence of the TolA binding region on ColE1-TR.

Though we cannot exclude the possibility that ColE1 T and R domain translocation depends on the C domain, we observe—as hypothesized by the pillar model—that colicin sits inside TolC but does not translocate. These results are in agreement with early studies of pore-forming colicins in which trypsin

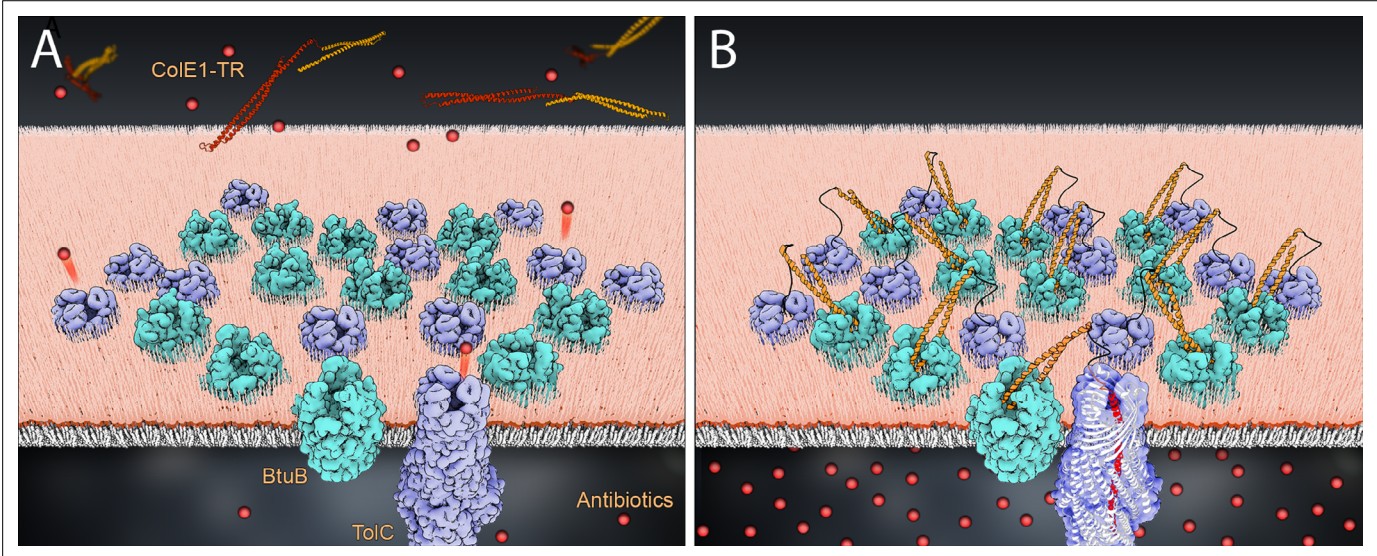

**Figure 6.** Model of colE1-TR inhibition of efflux. The R domain of ColE1-TR (orange) binds to BtuB (cyan) with high affinity and anchors colE1-TR to the surface of the cell. The T domain (red) then inserts into TolC (lavender), plugging the channel and blocking exit of antibiotics (*spheres*). We only see this interaction when ~20 colE1-TR bind in the same cluster though for simplicity only 12 BtuB and 12 TolC proteins are pictured here. (**A**) Before binding, antibiotics are effluxed. (**B**) After binding the colicin E1 fragments prevent some antibiotic efflux.

The online version of this article includes the following figure supplement(s) for figure 6:

**Figure supplement 1.** ColE1 TR sticks to the surface of *Escherichia coli* but the mechanism of C domain insertion is unclear.

added to the extracellular environment can reverse colicin activity (*Plate and Luria, 1972*; *Dankert et al., 1980*; *Dankert et al., 1982*; *Bénédetti et al., 1992*; *Duché, 2007*). We anticipate that the stable ColE-TR:TolC complex forms regardless of the presence of the C domain.

If the C domain does translocate while the T and R domains remain bound, we anticipate that the requirement for TolA is due to its role in membrane maintenance. Recruitment of the Tol-Pal system by a different group A colicin (ColE9) causes membrane defects similar to those in a Δ*tolA* strain (*Rassam et al., 2018*). Therefore, in the pillar model, colicin binding of TolA may facilitate C domain insertion through the defective membrane, whereas in the total thread model binding TolA transduces the proton motive force to unfold ColE1and pull it into the periplasm (*Figure 6—figure supplement 1*). Because we did not use the toxic C domain, our data cannot clarify what occurs after the T and R domain of full-length colicin bind the OMPs. Although our studies find that ColE1-TR remains on the surface of the bacteria, it is certain that the C domain, when present, translocates across the membrane into the target bacteria to kill the cell. Our data do not exclude the possibility that the translocation of the T and R domains occurs along with the C domain in full-length colicin, nor does it support T or R domain translocation. Thus, though our studies give a more complete view of the early stages of ColE1 binding the surface of its target, we cannot determine if insertion of the C domain occurs via the total thread model or via a modified pillar model.

Shortly after this work was made available, a similar structure demonstrated that KlebC, a rRNAse bacteriocin from *Klebsiella pneumoniae*, binds TolC from *Klebsiella quasipneumoniae* with a binding mode similar to that of ColE1 binding TolC (*Housden et al., 2021*). KlebC is closed in the unbound state as confirmed by the high-resolution X-ray crystal structure, which supports our proposed conformation (*Figure 2E*), and opens into a single-pass, kinked-helix to bind to TolC. Taken together, these findings suggest a conserved mechanism for bacteriocins that hijack trimeric efflux pump proteins.

In addition to demonstrating a colicin insertion mechanism, our observations form the basis of a means to manipulate bacterial import and efflux (*Figure 6*). Though colicin E1 confers some level of antibiotic potentiation, the relatively high concentration of ColE1-TR needed to inhibit efflux may be explained by a combination of the geometric constraints of creating large clusters of BtuB and TolC, and residually unblocked pore, even in the bound state as the large internal volume of TolC is not fully occupied by ColE1. Because of their limited potency, our ColE1-T and ColE1-TR fragments are not likely to be practical in direct applications of antibiotic potentiation. However, our proof-of-concept findings offer a potential roadmap for further development. A more potent TolC binder would not need the R domain anchor for affinity. Such a binder could be designed using the atomic details of the ColE1-T:TolC structure as a basis. Moreover, this scaffold can be used for efflux pump inhibitors for at least five other bacterial organisms. Each of these organisms has a structurally characterized outer membrane efflux pump that is homologous and structurally similar to TolC (*Franklin et al., 2018*).

More broadly, there are at least nine known OMP receptors for bacteriocins. These OMPs have a variety of functions including adhesion, iron transport, and general import (*Cascales et al., 2007*; *Kleanthous, 2010*). Using this same strategy with fragments derived from other bacteriocins may additionally allow for controlled inhibition of other bacterial functions.

# Materials and methods

**Key resources table**

| Reagent type (species) or resource | Designation | Source or reference | Identifiers | Additional information |
|---|---|---|---|---|
| Antibody | THE HisTag mAb Mouse (mouse monoclonal) | GenScript | Cat#A00186-100; RRID:19J001957 | (1:2000) |
| Antibody | Anti-SurA Rabbit polyclonal (rabbit polyclonal) | Cusabio | Cat#CSB-PA359693HA01ENV; RRID:E0109A | (1:2500) |
| Antibody | IRDye 800CW goat anti-mouse IgG secondary (goat polyclonal) | LI-COR | Cat#925-32210; RRID:C81106-01 | (1:10,000) |
| Antibody | IRDye 800CW Donkey anti-Rabbit (donkey monoclonal) | LI-COR | Cat#925-32213; RRID:C80829-10 | (1:10,000) |
| Cell line (*Escherichia coli*) | BW25113 | Coli Genetic Stock Center | Cat#7636 | |

*Continued on next page*

*Continued*

| Reagent type (species) or resource | Designation | Source or reference | Identifiers | Additional information |
|---|---|---|---|---|
| Cell line (*E. coli*) | JW5503-1 | Coli Genetic Stock Center | Cat#11430 | |
| Cell line (*E. coli*) | BL21(DE3) | Coli Genetic Stock Center | Cat#12504 | |
| Recombinant DNA reagent | pTrc99a-NoSS-TolC | This paper | Addgene: 155179 | Addgene: 155179 |
| Recombinant DNA reagent | pET303-colE1-T | This paper | Addgene: 155180 | Addgene: 155180 |
| Recombinant DNA reagent | pET303-colE1-TR | This paper | Addgene: 155181 | Addgene: 155181 |
| Recombinant DNA reagent | pET303-colE1-T-E192C | This paper | Addgene: 180233 | Addgene: 180233 |
| Recombinant DNA reagent | pET303-colE1-TR-E366C | This paper | Addgene: 180234 | Addgene: 180234 |
| Recombinant DNA reagent | pET303-colE1-TR-E366C-Δ1–40 | This paper | Addgene: 180235 | Addgene: 180235 |
| Recombinant DNA reagent | pET303-colE1-TR-Δ1–40 | This paper | Addgene: 180236 | Addgene: 180236 |
| Recombinant DNA reagent | pET303-colE1-T-P110A | This paper | Addgene: 180237 | Addgene: 180237 |
| Recombinant DNA reagent | MSP1D1 | Addgene | Addgene: 20061 | |
| Software, algorithm | cryoSPARC 2v2.12 | *Punjani et al., 2017* | https://cryosparc.com/ | |
| Software, algorithm | phenix v1.14-3260 | *Afonine et al., 2018* | https://www.phenix-online.org/ | |
| Software, algorithm | serialEM v3.7 | *Mastronarde, 2018* | https://bio3d.colorado.edu/SerialEM/ | |
| Software, algorithm | coot v0.8.9.2-re 7766 | *Emsley et al., 2010* | https://www2.mrc-lmb.cam.ac.uk/personal/pemsley/coot/ | |
| Software, algorithm | ISOLDE v1.0b4.dev1 | *Croll, 2018* | https://isolde.cimr.cam.ac.uk/static/isolde/doc/tools/ISOLDE.html | |
| Software, algorithm | UCSF ChimeraX | *Goddard et al., 2018* | https://www.cgl.ucsf.edu/chimerax/ | |
| Software, algorithm | PyMOL | Schrödinger, LLC | https://pymol.org/2/ | |

## *E. coli* strains

*E. coli* strains BW25113 and JW5503-1 were purchased from the Coli Genetic Stock Center (CGSC). JW5503-1 is a tolC732(del)::kan from the parent strain BW25113. BL21(DE3) was used for the production of the colicin constructs and TolC. BL21(DE3) has a premature stop codon at residue 58 of the *btuB* gene and therefore we used it as a Δ*btuB* strain for microscopy.

## Expression and purification

The gene for ColE1-TR was synthesized as a gBlock (Integrated DNA Technologies) and cloned into pET303. Inverse PCR was used to delete the R domain and produce colicin E1-T. Colicin E1-TR-GFP was produced by inserting GFP at the C terminus of ColE1-TR. A gBlock (Integrated DNA Technologies) was synthesized for GFP containing complementary flanking sequences to the plasmid with ColE1-TR and inserted with the omega-PCR technique. ColE1-TR$_{\Delta1–40}$ was produced by inverse PCR to delete residues corresponding to 1 through 40. All point mutants (ColE1-T-E192C, ColE1-TR-E366C, ColE1-TR-E366C$_{\Delta1–40}$, and ColE1-T-P110A) were generated by inverse PCR with mutagenic primers.

Plasmids were transformed into *E. coli* BL21(DE3) cells and plated on LB+ agar + 100 µg/ml carbenicillin. Single colonies were inoculated into 50 ml LB broth with 100 µg/ml carbenicillin and grown overnight at 37°C with shaking at 250 r.p.m. Proteins were produced by inoculating 1 L of TB supplemented with 0.4% glycerol, 10 mM MgCl$_2$, and 100 µg/ml carbenicillin with 20 ml of the overnight culture. The culture was grown at 37°C to an OD600 of 2.0 and induced with 1 mM IPTG. Expression cultures were then grown at 15°C for 24 hr and harvested at 4000*g* for 30 min at 4°C. Cell pellets were resuspended at 3 ml/g of cell pellet in lysis buffer (TBS, 5 mM MgCl$_2$, 10 mM imidazole, 1 mM PMSF, 10 µg/ml DNase, and 0.25 mg/ml lysozyme) and lysed via sonication (2 min, 2 s on, 8 s off, 40% amplitude, QSonica Q500 with 12.7 mm probe) in an ice bath. Lysates were centrifuged at 4000*g* for 10 min to remove un-lysed cells and debris. The supernatant was centrifuged again at 50,400*g* in a Beckman Coulter J2-21 for 1 hr at 4°C. Clarified lysates were applied to a 5-ml HisTrap FF column

and purified using an ÄKTA FPLC system with a 20-column volume wash step with binding buffer (TBS, 25 mM imidazole) and eluted using a linear gradient from 0% to 50% elution buffer (TBS, 500 mM imidazole) in 10 column volumes. Concentrated proteins were loaded onto a HiLoad Superdex 16/60 200 pg gel filtration column and eluted into phosphate-buffered saline (PBS) pH 7.4.

TolC production and purification were conducted in the same manner for preparation for cryoEM and for SEC. The gene for full-length TolC (a generous gift from R. Misra) was cloned into pTrc99a with the signal sequence deleted for expression into inclusion bodies. Plasmids were transformed into BL21(DE3) and plated on LB+ agar + 100 μg/ml carbenicillin. A single colony was picked and grown in LB at 37°C with shaking at 250 r.p.m. overnight. In the morning, 1 L of LB was inoculated with 20 ml of the overnight culture and grown at 37°C with shaking at 250 r.p.m. until the culture reached an OD600 of 0.6, at which point protein expression was induced with 1 mM IPTG for an additional 4 hr. Cells were then harvested at 4000$g$ for 30 min at 4°C. Cell pellets were resuspended in ml of lysis buffer (TBS, 5 mM MgCl$_2$, 0.2 mg/ml lysozyme, 5 μg/ml DNase, and 1 mM PMSF) at 3 ml/1 g of cell pellet and lysed via sonication (4 min, 2 s on, 8 s off, 40% amplitude, QSonica Q500 with 12.7 mm probe) in an ice bath. Cell lysates were centrifuged at 4000$g$ for 30 min at 4°C to harvest inclusion bodies. Inclusion body pellets were resuspended in inclusion body wash buffer (20 mM Tris pH 8.0, 0.5 mM EDTA, and 1% Triton X-100) and centrifuged at 4000$g$ for 30 min at 4°C. The inclusion body wash was repeated two more times. A final wash was done in 20 mM Tris pH 8.0 and inclusion bodies were stored at –20 °C. Inclusion bodies were solubilized in 20 mM Tris pH 8.0, 8 M urea at 500 μM. N-octyl-polyoxyethylene was added to 5 ml of solubilized TolC to a final concentration of 10% detergent and pipetted into a Slide-A-Lyzer G2 dialysis cassette with a 10,000 molecular weight cutoff. Refolding was initiated by dialysis in 5 L of 20 mM Tris pH 8.0, 100 mM NaCl at 4 °C with stirring overnight. TolC was centrifuged at 15,200$g$ for 10 min at 4°C to remove aggregates. The supernatant was filtered through a 0.22-μm filter, concentrated to 2 ml, applied onto a HiLoad 16/60 Superdex 200 pg column on an ÄKTA Pure FPLC system, and eluted with 1.5 column volumes in 20 mM Tris pH 8.0, 100 mM NaCl, 0.05% n-dodecyl-β-D-maltoside. TolC containing fractions were pooled and concentrated to 300 μM in Amicon centrifugal filters with molecular weight cutoff of 30 kDa.

MSP1D1 in pET28a was purchased from Addgene and was expressed and purified as previously described (*Hagn et al., 2018*).

## Peptide Synthesis

ColE1-T100-143 was synthesized using standard Fmoc chemistry with a CEM liberty blue microwave peptide synthesizer. The peptides were cleaved using a solution of 92.5:2.5:2.5:2.5 TFA:TIPS:H2O:-DoDt and the crude peptides were purified using HPLC. Analytical HPLC traces were acquired using an Agilent 1100 quaternary pump and a Hamilton PRP-1 (polystyrene-divinylbenzene) reverse phase analytical column (7 μm particle size, 4 mm×25 cm) with UV detection at 215 nm. The eluents were heated to 45°C to reduce separation of rotational isomers, and elution was achieved with gradients of water/acetonitrile (90:10 to 0:100 containing 0.1% TFA) over 20 min. Low-resolution mass spectra were obtained using a Waters Micromass ZQ 4000 instrument with ESI+ ionization.

## Extracellular protease digestion
### Growth, incubation with protein, and washing
5 ml overnight cultures of *E. coli* (BW25113) were grown in LB media at 37°C with shaking (250 r.p.m.) overnight. A 50 ml culture of LB was inoculated with 1 ml of the overnight culture and grown at 37°C with shaking (250 r.p.m.) until an OD600 of 0.6 was reached. The cultures were centrifuged at 4000$g$ at 4°C for 10 min. For the exogenous addition of protein, cell pellets were resuspended in cold PBS (Bio-Rad) containing 10 μM of either ColE1-T or ColE1-TR protein. Cells were incubated with protein with gentle rocking for 1 hr at 37°C. After incubation with protein, the cells were collected at 4000$g$ at 4°C for 10 min and resuspended in cold PBS without protein. The centrifugation and wash were then repeated a second time. The PBS washes are designed to remove unbound protein.

### Protease digestion of intact cells and lysed cells
Each sample is split into two 1-ml aliquots. The first aliquot is the intact cell condition which is placed on ice. The second aliquot is the lysed cells that control for protease activity. This aliquot was lysed by adding 0.25 mg/ml lysozyme (10 μl of a 25 mg/ml in 10 mM Tris pH 8.0, and 50% [v/v] glycerol) and

incubated at room temperature for 15 min followed by five freeze-thaw cycles in liquid nitrogen. The intact cells condition and lysed condition were both split into three 270-µl aliquots. For each triplet, trypsin was added at 5 and 20 µg/ml to digest any extracellularly exposed protein. 30 µl of PBS was added to the third aliquot of each triplet for a no trypsin sample. The three aliquots were incubated at room temperature to allow for trypsin digestion. After 30 min, 20 µl of 100 mM PMSF was added to terminate the trypsin digestion reaction. Then the samples associated with the intact cells condition were lysed. Lysozyme was added to a final concentration of 0.25 mg/ml and incubated at room temperature for 15 min followed by five freeze-thaw cycles in liquid nitrogen. All samples were stored at –20°C until ready for western blot analysis.

### Western blot

Samples were separated by SDS-PAG E. Coli cin proteins were probed with THE HisTag mAb mouse (GenScript). SurA was probed with anti-SurA Rabbit polyclonal (Cusabio). Western blots were imaged by fluorescence using an LI-COR Odyssey imager with secondary antibodies IRDye 800CW goat anti-mouse IgG and IRDye 800CW Donkey anti-Rabbit for Colicins and SurA, respectively.

## Single-molecule microscopy

Cysteine mutations were introduced at the C-terminus before the histidine tag for fluorophore conjugation. These constructs were purified as described in the Expression and purification section with the addition of 1 mM TCEP in all buffers. All subsequent steps were performed with limited exposure to light and in amber tubes. Cyanine3 (Cy3) maleimide (Lumiprobe) was reconstituted in DMSO. Fluorophore labeling was achieved by mixing a 20-fold molar excess of Cy3 maleimide to protein and incubating overnight at 4°C. Free dye was removed by gel filtration on a Sephadex NAP-10 G-25 column. Simultaneously to the dye removal, the sample was buffer exchanged into storage buffer (PBS pH 7.4, 1 mM DTT, and 1 mM EDTA). The degree of labeling was determined spectrophotometrically from the concentrations of the dye and protein solutions using their respective extinction coefficients, $\varepsilon$, as described by their manufacturers or for the proteins as estimated by Expasy ProtParam (Cy3 $\varepsilon_{548\ nm}$=162,000 L mol$^{-1}$ cm$^{-1}$; ColE1-T-E192C $\varepsilon_{280\ nm}$=9970 L mol$^{-1}$ cm$^{-1}$; ColE1-TR-E366C $\varepsilon_{280\ nm}$=14,440 L mol$^{-1}$ cm$^{-1}$). Labeling efficiencies were ~75% and ~85% for ColE1-T-E192C and ColE1-TR-E366C, respectively. Protein concentrations were adjusted according to the percentage of labeled protein.

Cultures of *E. coli* (WT, Δ*tolC*, or BL21(DE3)) were grown in LB medium at 37°C with shaking (180 r.p.m.) overnight, then transferred to MOPS minimal medium (Teknova) with 0.2% glycerol and 1.32 mM K$_2$HPO$_4$ and grown at 37°C for 13 hr. The sample was transferred to MOPS medium and grown to turbidity at 37°C overnight. A 1-ml aliquot of culture was centrifuged for 2 min at 4850*g* to pellet the cells. The pellet was washed in 1 ml MOPS and centrifuged a second time. The supernatant was then removed, and the cell pellet was resuspended in 500 µl MOPS. A 1.0-µl droplet of concentrated cells was placed onto a glass slide. Then, a 1.0-µl droplet of 1 µg/ml colicin E1 protein construct stock was added to the cells. The droplet was covered by an agarose pad (1% agarose in MOPS media) and a second coverslip.

Samples were imaged at room temperature using wide-field epifluorescence microscopy with sensitivity to detect single dye molecules as described previously (*Tuson and Biteen, 2015*). Briefly, fluorescence was excited by a 561-nm laser (Coherent Sapphire 560-50) for Cy3 or a 488-nm laser (Coherent Sapphire 488-50) for GFP. The lasers were operated at low power densities (1–2 W/cm$^2$), and fluorescence was imaged with an Olympus IX71 inverted microscope with a 100×, 1.40-NA phase-contrast oil-immersion objective and appropriate excitation, emission, and dichroic filters. A Photometrics Evolve electron multiplying charge-coupled device (EMCCD) camera with >90% quantum efficiency captured the images at a rate of 20 frames per second. Each detector pixel corresponds to a 49 nm×49 nm area of the sample. *E. coli* cells were identified and fluorescent signals were detected using the Single-Molecule Localization by Local Background Subtraction (SMALL-LABS) algorithm (*Isaacoff et al., 2019*).

## Co-elution

The interaction of TolC and colicin E1-T or -TR was determined by co-elution on an SEC column. Purified TolC and colicin E1-T or -TR were mixed at a 1:2 molar ratio to ensure an excess of colicin to saturate TolC binding sites and incubated at room temperature for 1 hr before loading onto a Superdex

200 Increase 10/300 GL column (GE Healthcare). The protein was eluted with 1.5 column volumes into 20 mM Tris pH 8.0, 40 mM NaCl, and 0.05% n-dodecyl-β-D-maltoside for ColE1-T. For ColE1-TR, the NaCl concentration was increased to 200 mM to prevent precipitation. Elution fractions were collected every 0.5 ml. Peak fractions were concentrated to 20 µl and analyzed by 4%–20% SDS-PAGE.

## Real-time efflux

Real-time efflux activity in the presence of ColE1-TR was determined as previously described with some modifications (*Bohnert et al., 2010*; *Bohnert et al., 2011*). Cells were resuspended to OD600 1.5 in cold PBS with and without 10–100 µM colicin proteins and incubated for 15 min on ice. To turn off efflux, 100 µM carbonyl cyanide m-chlorophenyl hydrazone (CCCP) was added. After an additional 15 min, the efflux dye NNN was added to the cells to 10 µM. To determine cell viability after treatment with CCCP and NNN, we counted the colony-forming units of treated and untreated cells. Treated cells had 93% viability compared to untreated cells as determined by colony-forming units (treated cells had 4.4×10$^9$ CFU/ml, untreated cells had 4.7×10$^9$ CFU/ml). The cells were incubated at 25°C with shaking at 140 r.p.m. for 2 hr. Cells were harvested at 3500*g* for 5 min and washed once in 20 mM potassium phosphate buffer pH 7 with 1 mM MgCl$_2$. Cell concentrations were adjusted to OD600 1.0 and placed on ice. Then, 2 ml of the cell suspension was loaded into a quartz cuvette (Agilent Technologies). Fluorescence was measured with an Agilent Cary Eclipse fluorescence spectrophotometer with slit widths at 5 and 10 nm for excitation wavelength of 370 nm and emission wavelength of 415 nm. Fluorescence measurements were taken every 1 s. After 100 s, 50 mM glucose was added to re-energize the cells and initiate efflux, and fluorescence data were collected for an additional 600 s. *Figure 5A–C* reflects time after glucose addition.

## Minimum inhibitory concentrations

MICs were determined using the broth dilution method (*Wiegand et al., 2008*) with some modifications in 96-well plates format using LB media in 100 µl well volumes. Cultures were grown at 37°C with shaking at 250 r.p.m. and OD600 was read on a Biotek plate reader after 20 hr. MICs are defined by the lowest concentration that prevents visible growth. We chose an OD600 of >0.1 as the cutoff for growth. We report MICs as the mean of 3 biological replicates with each replicate plotted (*Figure 5D*, *Figure 5—source data 1*). Due to the twofold discretized nature of concentration ranges used to determine MICs we do not report statistical significance values as is typical of MIC reporting.

## Reconstitution of TolC into amphipol

1 ml of TolC at 0.5 mg/ml was mixed with 0.75 ml of A8-35 (Anatrace) amphipol at 33 mg/ml for a 50-fold weight excess and incubated at room temperature for 30 min. Bio-Beads SM-2 resin that was washed in methanol and equilibrated with 20 mM Tris, 40 mM NaCl was added to the reaction mixture at 0.5 g/ml to initiate detergent exchange for A8-35 and incubated with rotation at 4°C overnight. The mixture was transferred to a tube with fresh Bio-Beads and incubated at 4°C for an additional 4 hr. The reaction mixture was loaded onto a HiLoad 16/60 Superdex 200 pg column on an ÄKTA Pure FPLC system and eluted with 1.5 column volumes in 20 mM Tris pH 8.0, 40 mM NaCl to remove free A8-35 and detergent. For colicin-bound TolC in A8-35, colicin E1-T was added to the reaction mixture at a >2 molar excess prior to gel filtration. TolC or colicin-bound TolC in A8-35 was concentrated to 2–4 mg/ml for cryoEM.

## Reconstitution of TolC into lipid nanodiscs

POPC (Avanti Polar Lipids) in chloroform was dried under a stream of nitrogen and freeze dried to remove residual chloroform. Lipids were reconstituted to 50 mM in cholate buffer (20 mM Tris pH 8.0, 100 mM NaCl, 0.5 mM EDTA, and 100 mM cholate) in an amber glass vial. The vial was submerged under a stream of warm water until the solution became clear. Lipids, membrane scaffold protein, and TolC were mixed in a 36:1:0.4 ratio as previously described (*Daury et al., 2016*). Final concentrations were 4.5 mM POPC, 125 µM MSP1D1, 50 µM TolC in a 2-mL reaction with cholate brought up to 20 mM and dodecyl-maltoside up to 0.1%. The reaction mixture was incubated on ice for 1 hr. Bio-Beads SM-2 resin that was washed in methanol and equilibrated with 20 mM Tris, 40 mM NaCl was added to the reaction mixture at 0.5 g/ml to initiate nanodisc formation and incubated with rotation at 4°C overnight. The mixture was transferred to a tube with fresh Bio-Beads and incubated at 4°C for

an additional 4 hr. The reaction mixture was loaded onto a HiLoad 16/60 Superdex 200 pg column on an ÄKTA Pure FPLC system and eluted with 1.5 column volumes in 20 mM Tris pH 8.0, 40 mM NaCl to separate TolC inserted into nanodiscs from empty nanodiscs. For colicin-bound TolC in nanodiscs, colicin E1-T was added to the reaction mixture at a >2 molar excess prior to gel filtration. TolC or colicin-bound TolC in nanodiscs was concentrated to 2–4 mg/ml for cryoEM.

## Cryo-electron microscopy

3 µl of protein solution (TolC/ColE1-T in amphipol or TolC/ColE1-T in nanodiscs) was diluted to approximately 1.05 mg/ml concentration, applied to a glow-discharged TEM grid, and plunge-frozen in ethane using a Vitrobot Mark IV (FEI Company) with grade 595 filter paper (Ted Pella). Glow discharge was performed in ambient atmosphere at 0.39 mBar pressure. Imaging was performed using a Talos Arctica (FEI Company) operated at 200 kV with energy filter and K2 Summit (Gatan, Inc) for detection. To collect multiple images per hole while maintaining parallel illumination conditions, a nonstandard 20 µm condenser aperture was used to image TolC-ColE1-T in nanodiscs. At nominal magnification of 205,000×, images were acquired in counting mode with a calibrated pixel size of 0.6578 Å. Fresnel fringes attributable to the beam interaction with the aperture were often seen in images. Some investigators have moved the microscope stage and altered the nominal objective lens true focus point to generate a fringe-free condition (*Weis and Hagen, 2020*). In this study, imaging at 205,000× with a 20-µm aperture yielded better results than imaging at 130,000× with a 50-µm aperture; at 130,000× with a 20-µm aperture, the fringes were extremely severe due to the larger field of view, so a full dataset was not collected with those conditions. TolC in nanodiscs (without ColE1-T) was imaged at 130,000× with a 50-µm condenser aperture (*Figure 1—source data 1*).

Micrographs were collected with SerialEM (*Mastronarde, 2018*) using in-house modifications of the scripts of Chen Xu (sphinx-emdocs.readthedocs.io). Briefly, multishot imaging was configured with 4 images per hole for each of 16 holes; intermediate-magnification montages of grid squares were acquired; points were selected manually for collection of 64 images per point; images were acquired using coma-compensated image shift as gain- and dark-corrected LZW-compressed TIFs. Side, top, and oblique views were seen in areas of thin ice. During screening, ice thickness was estimated at 17–30 nm by the method of $I_0/I_{ZLP}$ (*Rice et al., 2018*).

The collection of micrographs of TolC without colicin at 130,000× magnification has been previously described (*Budiardjo et al., 2021*).

## 3D reconstruction and modeling

Final reconstructions were obtained using cryoSPARC 2 (*Punjani et al., 2017*). 1018 micrographs were collected of amphipol-embedded TolC/ColE1-T. Micrographs were motion-corrected using RELION 3 (*Zivanov et al., 2018*). CTF parameters were determined by means of *ctffind* (*Rohou and Grigorieff, 2015*). 115,362 particles were selected with crYOLO (*Wagner et al., 2019*). 2D classification revealed that many particles had aberrant morphology and only 24,624 (21%) were selected for 3D reconstruction. Ab initio reconstruction in cryoSPARC 2 (*Punjani et al., 2017*) was effective at recovering a map whose shape was similar to that of previously described TolC trimers. However, the data could only be refined to a nominal global resolution of 6.0 Å, and lumenal density was insufficiently resolved. 4,492 micrographs were then collected of nanodisc-embedded TolC/ColE1-T and processed similarly. Of the 339,779 particles detected by cryoSPARC Live, 179,834 (53%) were in good classes. Although there were slightly fewer particles per micrograph in the nanodisc data set, more particles per micrograph were usable. Beginning with the *ab initio* model and mask derived from the amphipol data, this particle set was refined by cryoSPARC two non-uniform refinement with or without imposed $C_3$ symmetry. The maps refined to nominal global resolutions of 2.81 Å and 3.09 Å for the symmetrized and asymmetric maps, respectively. There was local variation in resolution within the map, with consistent, high resolution in the middle of TolC and lower resolution at the lids and for ColE1-T. Local resolution was computed in cryoSPARC by the locally-windowed FSC method (*Cardone et al., 2013*) and rendered with UCSF Chimera. To reduce the voxel-based values to averages for four regions of the complex, the local resolution map was masked to include only voxels within 3 Å of a modeled atom and then the median value was calculated for those voxels closest to ColE1, closest to TolC residues 168–172 and 386–390, closest to TolC residues 285–301 and 76–82, or closest to other TolC residues. Furthermore, it is notable that while the nanodisc appears as a double-belt in the symmetrized map,

in the asymmetric map the nanodisc protein mostly appears on the side of TolC that is bound to ColE1-T. One possible explanation is that, despite masking, nanodisc asymmetry is a confounder of the asymmetric refinement and is one source of heterogeneity in the data. Another possibility is that the C-terminus of ColE1-T forms an interaction with the nanodisc, causing preferential alignment of the nanodisc with respect to the TolC/ColE1-T complex.

196,158 particles of TolC without colicin were obtained as previously described (*Budiardjo et al., 2021*). Homogeneous refinement yielded a structure at 2.89 Å; local motion correction and global CTF refinement yielded a final map at 2.84 Å.

Modeling was initiated by rigidly docking a crystal structure of TolC in complex with hexamminecobalt (1TQQ) (*Higgins et al., 2004*) into the symmetrized map density. Automated, semi-automated, and manual real-space refinement was performed using phenix (*Afonine et al., 2018*), ISOLDE (*Croll, 2018*), and coot (*Emsley et al., 2010*). For TolC with ColE1-T, additional refinement was performed in AMBER using the cryoEM density map as a restraint.

Although additional residues are present at the TolC C-terminus, these were not modeled because the density becomes unsharp after residue 428. Blurred density in the map suggests that the C-terminus follows helix three toward the periplasmic end of the molecule. ColE1-T was modeled *ab initio*. The threefold-symmetrized map contains density at ~1/3rd occupancy for ColE1-T and this density contains some high-resolution information not present in the asymmetric map, except near the TolC lid regions where symmetrization overlays ColE1-T density with TolC density at a threefold-related position. First, polyalanine helices were placed in the helical density in coot. Cross-correlation coefficients for both helices are higher with the N-terminus oriented toward the periplasm. Moreover, the side chains angle slightly toward the N-terminus like the fronds of an evergreen angle toward the ground in typical 'Christmas tree' conformation, indicating that this is the correct chain orientation. An estimate of the register was made by visual inspection of potential anchor residues. Finally, the hinge region was filled in using phenix and coot. This completed chain was refined against the asymmetric map in ISOLDE. Iteration between phenix, coot, and ISOLDE was continued until acceptable fit to density was achieved. In the case of TolC with ColE1-T, the map was further improved by combining map information with molecular dynamics force fields (*Wu et al., 2013*). Briefly, starting with the phenix/coot/ISOLDE-refined model, we performed restrained simulated annealing in AMBER, heating from 0K to 300K for 0.2 ns, holding constant temperature for 1 ns, and then cooling to 0K over 0.2 ns. The cryoEM density map is utilized as a restraint potential in the annealing so that both map information and AMBER force field information are simultaneously utilized to obtain an optimum model consistent with the data (*Wu et al., 2013*). The protein force field used the ff14SB force field (*Maier et al., 2015*) and a generalized Born implicit solvent model with $igb$=8 (*Nguyen et al., 2013*), and a nonbonded cutoff of 20 Å. The relative weight of real-space map-based restraints and the force field was fixed using $fcons$=0.02. For ColE1-T, information from the symmetrized map was integrated into the modeling procedure during manual remodeling in coot, but map-based refinement in phenix, ISOLDE, and AMBER was against the asymmetric map. TolC without ColE1-T was modeled similarly but using the TolC-ColE1-T structure as a starting point instead of 1TQQ, and without final AMBER refinement.

Molecular representations were generated with Chimera, ChimeraX (*Goddard et al., 2018*), or PyMOL (Schrödinger, LLC).

## AlphaFold model of ColE1

AlphaFold (*Jumper et al., 2021*) was used to generate a predicted structural model for full-length ColE1. The amino acid sequence for residues 1–522 of ColE1 was submitted to the AlphaFold Colab (https://colab.research.google.com/github/deepmind/alphafold/blob/main/notebooks/AlphaFold.ipynb) with the 'prokaryotic' parameter set as 'True'.

## Acknowledgements

The authors gratefully acknowledge Daniel Montezano, Pinakin Sukthankar, Rik Dhar, Dwight Deay, Scott Lovell, Matthias Wolf, Alexander Little, Heather Shinogle, and Sarah Noga for discussions and feedback, Mark Richter for the use of his fluorometer, Rajeev Misra for the pTrc vector containing the TolC gene, Vasileios Petrou for guidance on nanodiscs, Chamani Perera for peptide synthesis, Karen Marom for editorial guidance, Batika Saxena for artistic contributions, Joseph Lubin and William

Hansen for introductions, and the Office of Advanced Research Computing (OARC) at Rutgers for high-performance computing resources. Funding: NIH award R21-GM128022 to JSB, NIGMS awards DP2GM128201, P20GM113117, P20GM103638, and the Gordon and Betty Moore Inventor Fellowship to JSGS, NIGMS awards P20 GM103418 and 2K12GM063651 to SJB.

## Additional information

### Funding

| Funder | Grant reference number | Author |
|---|---|---|
| National Institute of General Medical Sciences | DP2GM128201 | Joanna SG Slusky |
| National Institute of General Medical Sciences | P20GM113117 | Joanna SG Slusky |
| National Institute of General Medical Sciences | P20GM103638 | Joanna SG Slusky |
| Gordon and Betty Moore Foundation | Moore Inventor Fellowship | Joanna SG Slusky |
| National Institute of General Medical Sciences | P20 GM103418 | S Jimmy Budiardjo |
| National Institute of General Medical Sciences | 2K12GM063651 | S Jimmy Budiardjo |
| National Institute of General Medical Sciences | R21-GM128022 | Julie S Biteen |

The funders had no role in study design, data collection and interpretation, or the decision to submit the work for publication.

### Author contributions

S Jimmy Budiardjo, Conceptualization, Formal analysis, Funding acquisition, Investigation, Methodology, Project administration, Writing – original draft, Writing – review and editing; Jacqueline J Stevens, Anna L Calkins, Emre Firlar, Formal analysis, Investigation; Ayotunde P Ikujuni, David A Case, Formal analysis, Investigation, Methodology; Virangika K Wimalasena, Investigation; Julie S Biteen, Formal analysis, Funding acquisition, Methodology, Supervision, Writing – original draft, Writing – review and editing; Jason T Kaelber, Formal analysis, Investigation, Methodology, Supervision, Writing – original draft, Writing – review and editing; Joanna SG Slusky, Conceptualization, Formal analysis, Funding acquisition, Investigation, Project administration, Supervision, Writing – original draft, Writing – review and editing

### Author ORCIDs

S Jimmy Budiardjo http://orcid.org/0000-0003-2094-9179
Jacqueline J Stevens http://orcid.org/0000-0003-2235-0522
Ayotunde P Ikujuni http://orcid.org/0000-0001-8951-3440
Virangika K Wimalasena http://orcid.org/0000-0002-1061-3439
Emre Firlar http://orcid.org/0000-0003-0190-6528
Julie S Biteen http://orcid.org/0000-0003-2038-6484
Jason T Kaelber http://orcid.org/0000-0001-9426-1030
Joanna SG Slusky http://orcid.org/0000-0003-0842-6340

### Decision letter and Author response

Decision letter https://doi.org/10.7554/eLife.73297.sa1
Author response https://doi.org/10.7554/eLife.73297.sa2

## Additional files

### Supplementary files
• Transparent reporting form

### Data availability

CryoEM maps and models have been deposited with accession codes EMD-21960, EMD-21959, PDB ID 6WXI, and PDB ID 6WXH. The following data are publically available for the two structures: 6WXH TolC + colE1 Structure: https://files.rcsb.org/download/6WXH.cif EM map: https://files.rcsb.org/download/6WXI.cif Validation report: https://ftp.wwpdb.org/pub/emdb/structures/EMD-21959/map/emd_21959.map.gz; 6WXI TolC alone Structure: https://files.rcsb.org/download/6WXI.cif EM map: https://files.rcsb.org/pub/pdb/validation_reports/wx/6wxh/6wxh_full_validation.pdf Validation report: https://files.rcsb.org/pub/pdb/validation_reports/wx/6wxi/6wxi_full_validation.pdf.

The following datasets were generated:

| Author(s) | Year | Dataset title | Dataset URL | Database and Identifier |
|---|---|---|---|---|
| Kaelber JT, Budiardjo SJ, Firlar E, Ikujuni AP, Slusky JSG | 2021 | Colicin E1 fragment in nanodisc-embedded TolC | https://www.ebi.ac.uk/emdb/EMD-21960 | EBI, EMD-21960 |
| Kaelber JT, Budiardjo SJ, Firlar E, Ikujuni AP, Slusky JSG | 2021 | Colicin E1 fragment in nanodisc-embedded TolC | https://www.rcsb.org/structure/6WXI | RCSB Protein Data Bank, 6WXI |
| Kaelber JT, Budiardjo SJ, Firlar E, Ikujuni AP, Slusky JSG | 2021 | Colicin E1 fragment in nanodisc-embedded TolC | https://www.ebi.ac.uk/emdb/EMD-21959 | EMDB, EMD-21959 |
| Kaelber JT, Budiardjo SJ, Firlar E, Ikujuni AP, Slusky JSG | 2021 | Colicin E1 fragment in nanodisc-embedded TolC | https://www.rcsb.org/structure/6WXH | RCSB Protein Data Bank, 6WXH |

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
