## [Editor Report]

Colicins are plasmid-encoded toxins produced by bacteria to kill closely related bacteria that compete for scarce resources. While the individual proteins used for binding and taking up colicins in target bacteria are well-known, the precise pathway that those toxins ultimately take to make it into the cell is unclear. Here the authors reveal the cryo EM structure of the outermembrane protein TolC in complex with colicin E1 toxin to shed new mechanistic insights into this toxin import mechanism.

---

## [Decision Letter]

**Decision letter after peer review:**

Thank you for submitting your article "Colicin E1 opens its hinge to plug TolC" for consideration by *eLife*. Your article has been reviewed by 3 peer reviewers, including David Drew as the Reviewing Editor and Reviewer #1, and the evaluation has been overseen José Faraldo-Gómez as the Senior Editor. The following individuals involved in review of your submission have agreed to reveal their identity: Bert van den Berg (Reviewer #2); Karen Jakes (Reviewer #3).

Essential revisions:

1. Membrane localization experiments with ColE1-T and ColE1-TR are described in Figure 4. However, more experimental details need to be included in the figure legends and Methods, especially regarding protein concentrations in the experiments shown in Figure 4. For example, what is analysed in intact cells? Is bound toxin washed off into the supernatant? If the toxin has very high affinity will it come off the OMP receptor(s)? The relative concentrations of the colicin constructs used is not given, so there's no way to easily interpret some of the data. Furthermore, Figure 4C shows one cell each, and one could argue that only the best data is shown involved. Could the authors (perhaps as source data) show panels with many cells to show any differences between strains more objectively?

2. The Discussion section needs to better describe the differences between the "pillar" and "total-thread" models, perhaps by also including a schematic of both. Furthermore, the experimental evidence in this paper supports the total-thread model, whereas the more ambiguous experiments (trypsin digestion) are only suggestive of the pillar model. Please make the final proposed model clearer in the discussion and include how the downstream pore-forming cytotoxic domain is translocated.

3. Experiments with GFP attached to colE1-TR suggest efficient uptake in a tolC-dependent process. What would be a very informative further would be to delete the TolA box from the N terminus of the T domain of the T-R construct and determine whether uptake still occurs. If so, that would argue further against a translocation model whereby the C-terminal channel-forming domain of colicin E1 slithers down the outside of TolC, with its C terminus first, as proposed by Cramer et al. (2018). Additional control would be to mutate the hinge-proline.

4. The Housden klebicin paper was published very recently describing detailed studies of the interaction of TolC with another bacteriocin, including cryo-EM structures (Housden et al., Nature Commun 12(1):4625). While the work described here has been carried out independently, some acknowledgement of this work in the discussion is warranted.

*Reviewer #1:*

Colicin E1 is an *E. coli* bacteriocin that hijacks the outer membrane proteins TolC and BtuB to enter the cell. Here the authors reveal the cryo EM structure of TolC in complex with colicin E1. They show how Colicin E1 structure rearranges from a helical hairpin into an single, long extended helix that plugs the TolC pore. They further show that *E. coli* cells that have TolC proteins plugged with a Colicin E1 fragments are more susceptible to antibiotics. Taken together, they have outlined the structural basis for Colicin E1 recognition and shown the applicability of this approach for bacterial pathogenicity.

Overall, I can find little technical issues with the paper and I think the conclusions drawn from the experiments are sound. The paper is well-written and the figures are easy to follow. From a mechanistic viewpoint the major findings was that it was unexpected for colicin E1 to open up from a helical hairpin into a single helix inside TolC. While the author have succeeded to understanding how the toxin engages with its receptor its still unclear how the downstream pore-forming cytotoxic domain is translocated across TolC.

*Reviewer #2:*

The paper by Slusky and colleagues presents cryo-EM structures of *E. coli* TolC in the absence and presence of the N-terminal translocation domain of Colicin-E1 (ColE1-T). Via SEC they demonstrate that the toxin domain binds tightly to TolC. Interestingly, the structure shows that the colicin is bound inside the long TolC barrel, with most of the colicin-OMP contacts provided by the β-barrel part of TolC. The cryo-EM data are nice and the map of the complex is of very good quality and supports the ColE1-T model built by the authors. While the involvement of TolC (and another OMP named BtuB) in ColE1 cell entry was known, the structure provides in my opinion clear support for a model in which the entire ColE1 is transported into the cell via the TolC conduit, utilising the proton motive force to power translocation via TolA. This latter part is not well described by the authors and I find the discussion of the two models not very clear and too much in favor of the "pillar" model (see detailed comments; I think also a schematic figure is warranted showing the two models). Another major issue that needs addressing is the fact that the authors have missed (or ignored) a very similar paper that was published very recently and which describes detailed studies of the interaction of TolC with another bacteriocin (Klebicin), including cryo-EM structures (Housden et al., Nature Commun 12(1):4625). Thus, in contrast to what is claimed by the authors, theirs is not the first structure of a bacteriocin bound to TolC. The authors should compare their structure with that of Housden et al. (which is very similar) and rewrite their discussion based on both papers.

To support their structural data the authors also carried out membrane localization experiments with ColE1-T and ColE1-TR (includes the receptor binding domain that interacts with the primary receptor BtuB), described in Figure 4. I didn't find this very clear. In particular I'm not sure what is analysed in Figure 4B? What do you analyse in intact cells? Bound toxin that is washed off and ends up in the sup? If so, what if the toxin has very high affinity and won't come off the OMP receptor(s)? The methods section is not complete and, rather than refer to a previous paper, the procedures should be fully described here. Figure 4C shows one cell each, and one could argue that cherry picking might be involved. Could the authors (perhaps as source data) show panels with many cells to show any differences between strains more objectively?

The final data provided by the authors utilises the fact that TolC is part of the AcrAB-TolC RND efflux pump that moves noxious substances including antibiotics out of the cell. They hypothesise that, when ColE1-T or ColE1-TR is bound to TolC (plugging the efflux channel), the pump should not work or work less efficiently. Indeed, MIC values for several antibiotics are lower in the presence of certain toxin fragments, showing that cells are sensitised due to reduced drug efflux. The required concentration of the colicins are high, however, suggesting that the pump is still partly functional while the colicin is bound, a notion supported by the structure. Together, these data lead to the interesting hypothesis that modified bacteriocins that hijack functionally important OMPs might be used to potentiate antibiotics. Together, this paper provides a nice advance in our thinking on how bacteriocins gain entry into cells (but the discussion should be improved substantially). It is also consistent with the more detailed study by Housden et al., which is satisfying. The potential application towards potentiating antibiotics via colicin-based targeting of OMPs like TolC could be an important future area of research.

*Reviewer #3:*

Colicins are plasmid-encoded toxins produced by bacteria to kill closely related bacteria that compete for scarce resources. While the individual proteins in the target bacteria used for binding and uptake of colicins are well-known, there remains some controversy about the precise pathway that those toxins ultimately take to reach the target of their ultimate lethal act. In this work by Budiardjo et al., the path for colicin E1, which ultimately kills target *E. coli* by making an ion-permeable channel in their inner membrane, is clarified.

All of the E colicins use the vitamin B12 receptor, BtuB, as their primary outer membrane receptor to concentrate the colicin at the cell surface via an interaction with the central receptor-binding (R) domain. The enzymatic E colicins, E2-E9, then use the trimeric porin, OmpF, and its interactions with the N-terminal translocation (T) domain of the colicin to thread through one of the three pores of OmpF to ultimately bring the catalytic, carboxy-terminal end of the colicin into the periplasm from which it ultimately reaches and crosses the inner membrane. That process has been very thoroughly examined and documented, principally by the work of Kleanthous, Housden and their co-workers, as cited in references in this manuscript and others cited in this review. Colicin E1, which kills target bacteria by making a channel in their inner membrane and thereby depolarizing the cell, also uses BtuB as its primary outer membrane receptor. But then it differs from the other E colicins in hijacking the TolC channel, whose normal role in the cell is in drug and toxin export, and uses that channel as a means to enter the target cell. The isolated colicin E1 T domain has been found to exist in solution as a helical hairpin (Zakharov et al., 2016), but whether in vivo binding to TolC occurs as that same hairpin, with both ends pointing outside the target bacterium, or whether the hairpin unfolds, possibly around a TolC box previously shown to be essential for cytotoxicity, as proposed (Jakes, 2017), was not known. In this work, it is now clearly shown by cryoEM that the T domain hairpin unfolds completely and is captured inserted inside TolC. The previously identified TolC box acts as a hinge with a kink at a central proline in the 21-residue box sequence (residues 100-120). Residues in the TolC box are involved in specific interactions with residues of TolC within the channel. The interaction with the T domain peptide alone was sufficient to dilate both the periplasmic and extracellular apertures of TolC, without involvement of any other proteins or motive force, a very signficant finding. The structure and the specific residue interactions elucidated in this work help explain earlier results with mutations in TolC that block killing by colicin E1 (ref. #37). The structure reported here negates earlier arguments against models in which colicin E1 threads through TolC to enter the periplasm. The direction of insertion is with the N terminus facing the periplasmic end of TolC. This arrangement would logically be expected, since the known TolA box sequence lies very near the N terminus of the colicin E1 T domain and needs to interact in the periplasm with TolA, a bacterial protein anchored in the inner membrane which extends into the periplasm, is thought to provide the energy necessary to mobilize translocation of the colicin and has been demonstrated genetically to be absolutely required for colicin lethality.

Budiardjo et al. have failed to acknowledge the recent publication of nearly identical results with the closely-related *Klebsiella* pneumoniae klebicinC uptake via TolC (Housden et al., 2021, Nat. Commun. 12(1): 4625), in which the klebicinC T domain was also captured inside the channel of the K. pneumoniae TolC channel. In those experiments, the stretch of T domain bound inside of TolC was protected from digestion by trypsin, with residues at both ends protruding from TolC subject to protease digestion. Thus, these two different bacteriocins, targeting different bacteria, both insert their N-terminal translocation domains into the TolC protein of their respective targets.

A second very significant result from this manuscript is that binding of colicin E1 T domain inhibits drug efflux and thereby reduces the minimum concentration of some drugs to effectively kill bacteria. A significant effect was only seen in constructs that also included the R domain of the colicin, which may simply be serving to increase the concentration of T domain at the cell surface and thereby making its interaction with TolC more efficient. A shortcoming of these experiments is that the relative concentrations of the colicin constructs used is not given, so there's no way to interpret the lack of much effect from T domain alone. But the basic result suggests that there may be a route to using the E1 T domain as a means to reduce resistance of pathogenic bacteria to standard antibiotics, as has been suggested previously (Jakes, 2017).

In vivo experiments to determine whether the T domain or T-R domain were protected from trypsin digestion by virtue of having been internalized by *E. coli* are difficult to interpret. The protocol cited in the manuscript is a general protocol, and the relative amounts of protein used are not given. It is perhaps unsurprising that no T domain is seen bound to either intact or lysed cells, while T-R is detected, since the binding of isolated T domain, via TolC, has been shown to be orders of magnitude less efficient than that of R domain to BtuB (Jakes, 2017). The authors point out that the proteolysis results may just be a reflection of a small number of molecules entering the cells. Similarly, the failure to see much transport of the fluorescently-labeled colE1-TR-Cy3 may also be a reflection of a combination of inefficient binding and the bulky Cy3 label impeding passage through TolC. In fact, recent work on transport of colicin E9 through OmpF suggests that relatively small differences in the size of fluorescent labels attached to colicin domains can have a measurable effect on uptake of the labeled proteins (Francis et al. EMBO J. e108610/2021), and the labels used in that work had lower molecular weights than the Cy3 used in the experiments described here. Experiments with GFP attached to colE1-TR suggest more efficient uptake in a tolC-dependent process. What would be a very informative further experiment would be to delete the TolA box from the N terminus of the T domain of the T-R construct and determine whether uptake still occurs. If so, that would argue further against a translocation model whereby the C-terminal channel-forming domain of colicin E1 slithers down the outside of TolC, with its C terminus first, as proposed by Cramer et al. (2018).

[Editors' note: further revisions were suggested prior to acceptance, as described below.]

Thank you for resubmitting your work entitled "Colicin E1 opens its hinge to plug TolC" for further consideration by *eLife*. Your revised article has been evaluated by José Faraldo-Gómez (Senior Editor) and a Reviewing Editor.

The manuscript has been improved but there are some remaining issues that need to be addressed, as outlined below:

1. While we realise the PDBs were released earlier than those of Housden et al. it would be better to remove the instances of "first" in the text as, after all, they are not really needed.

2. The modified text for points 23 and 24 in the previous submission was still unclear to Referee 2. We would ask you to try to re-address these points (re-pasted below for clarity).

23. line 253: "need to be in close proximity for colE1 binding to occur" needs a reference.

24. line 254: "when BtuB clusters together in groups of 12 or more, it may exclude TolC" needs a reference. Also, please explain "These cluster geometries would therefore lower the number of full binding sites available for the T and R domains of colicin E1."

3. Throughout the manuscript, reference is made to the TolA box of colicins E3 and E9. However, neither of those colicins actually has a TolA box. Beginning with the early work of Bouveret et al. (Mol. Micro 1998 27, 143-157) and subsequently by others, it has been shown that what had been called the TolA box in earlier work is actually a TolB box. Colicins E2-E9 and A all interact with TolA indirectly, by binding to TolB in the periplasm, and TolB then engages TolA. The fullest picture of those interactions is in the recent work of Francis et al. (2021), which is cited in this manuscript. The only E colicin known to have a TolA box is E1. A reference (probably Pilsl and Braun, 1995) is needed when referring to the E1 TolA box. All references to a TolA box in colicins E3 or E9 need to be fixed.

4. While the experimental details and source data have been expanded and clarified, the implications of both the trypsin digestion experiments and fluorescence labeling are less clear than the cryo-EM and drug efflux experiments, and some caveats should be added to the discussions of those experiments.

5. The argument between the two competing models may require some further clarification. Referee 3 has made some suggestions below. Please look at these and amend appropriately as you see fit.

Specific comments and corrections:

Line 62: …targets and killing mechanisms. Most colicins…

Lines 69-71 should read: …of ColE3 or ColE9 initiates translocation using the secondary OMP receptor/translocator, OmpF, to access the Tol-Pal system in the periplasm. For most colicins, the T domain requires an OMP distinct from the R domain target for colicin translocation.

Line 72-73: …structurally characterized bound to their OMP primary receptors. (Kurisu…, Sharma…, Buchanan et al., 2010). Those structures all reveal that the plugged β-barrels remain plugged upon colicin binding, so another route through the outer membrane is required.

Line 75-76 ff: …that T domain can pass fully through one pore of its membrane translocator, OmpF, and loops around to insertv in a second pore of the trimer from the periplasmic side, while the R domain initially interacts with…its receptor, BtuB. Those same studies and earlier ones clearly demonstrated that full engagement of the E9 T domain with its TolB translocation partner in the periplasm results in disengagement of the R domain from BtuB (Francis et al. 2021 and references therein). Experiments such as these and those that preceded this most recent work are the basis for the "total thread" model for colicin translocation, in which the colicin is pulled through the same pore of OmpF through which its T domain initially entered. The process is energized by the TolABQR system.

Line 82: death. Unlike any of the other E colicins, the T domain of colicin E1…

Line 83: …and the R domain binds BtuB, the primary receptor shared by all E colicins.

Line 98: …and disrupt TolC channel conductance in vitro, in planar lipid bilayer membranes…

Line 101: …in solution similar to another colicin T domain from colicin Ia, one that does not interact with TolC (Wiener et al., 1997).

Line 223: …these residues are contact sites…

Line 290: Should read: Given the two-dimensional surface…

Line 370: Add Francis et al., 2021 reference here.

Line 376: Why was the hinge opening unanticipated?

Line 378: …as the total thread model hypothesized and consistent with experimental data that support this model for colicin E9 (Housden et al., 2005; Housden et al., 2013, Francis et al., 2021).

Line 380: …clustered and does not diffuse…

*Reviewer #1:*

I am satisfied with the revised submission and have no further concerns.

*Reviewer #2:*

The revised manuscript by Slusky has substantially improved based on the reviewer comments and in particular it is now much clearer. I have just a few remaining minor issues for consideration:

point 2: I understand the reasoning of the authors. However, I'm not sure if it is entirely appropriate to allow priority claims based on non peer-reviewed preprints. While I do realise the PDBs were released earlier than those of Housden et al., in my opinion it would be better to remove the instances of "first" (as indeed is the norm in some other journals) in the text as, after all, they are not really needed. I don't know what the policy is of *eLife* in this regard, and leave it up to the editor to decide.

points 23 and 24: unfortunately the modified text is still unclear to me.

*Reviewer #3:*

The authors have very thoroughly addressed all of the reviewers' comments, provided additional experiments and methodological descriptions, and fixed the reference list to remove duplications and make it a more user-friendly format.

From that perspective, I should say that the paper is acceptable now. But in the process of expanding parts of the Introduction and Discussion, I feel they have done their work and the field a disservice by giving "first billing" to a model for which there is scant experimental support and giving short shrift to the model for which there is now extensive experimental support and which common sense should support.

In their expanded Introduction, the authors once again lead with a lengthy elaboration of the "pillar model" for the translocation of the colicin E1 pore-forming domain across the outer membrane of target *E. coli*. That model was introduced simply because of the assumption by Zakharov et al. (2016) that the helical hairpin structure of the E1 T domain observed in solution from CD spectra is maintained when the T domain binds in vitro to TolC, as it was shown to do by SEC, occlusion of TolC channels in planar lipid bilayer membranes (Zakharov et al., 2016), and in vivo by protecting sensitive *E. coli* from killing by active colicin E1 (Jakes, 2017). On the contrary, the current work has clearly demonstrated that the hairpin unfolds and inserts, amino-terminus first, into TolC., so the whole basis for a model with both N- and C-termini pointing to the outside of the target cell is gone. (In addition, recent work on the Klebicin C bacteriocin has shown that locking that protein's T domain helix in its closed state, with an inserted disulfide bond, kills its activity in the oxidized state, but not when it's reduced (Housden et al., 2021)). Another argument made by Zakharov et al. against insertion of the T domain into TolC was that the periplasmic aperture of the TolC channel is too narrow for a polypeptide to exit, so therefore the colicin couldn't possibly thread into the periplasm via that route. The structure shown in Figure 1B demonstrates that the binding of T domain within the channel opens that aperture; the colicin domain is easily accommodated within TolC. Thus, the colicin polypeptide or a part of it could transit through TolC.

The sections on potentiation of antibiotics by the E1 T domain are additional evidence that the colicin domain is occupying TolC and inhibiting its normal role in drug efflux. It's likely that the failure to see much of an effect with T domain, as compared to R-T, is simply the result of the much tighter binding of R to BtuB, which then concentrates T domain at the cell surface and makes it more likely to find and enter a TolC channel.

While the experimental details and source data have been expanded and clarified, the implications of both the trypsin digestion experiments and fluorescence labeling are less clear than the cryo-EM and drug efflux experiments, and some caveats should be added to the discussions of those experiments. It's not clear what not finding protected T domain tells us. Recently published experiments with the related KlebC and the K. quasipneumoniae TolC (Housden et al. 2021) clearly show that a significant segment of the KlebC T domain is protected from trypsin digestion when bound and digested in vitro. Mass spec analysis of the protease digestion products showed that the KlebC T domain extends out of the TolC channel at both the periplasmic and extracellular ends, and those parts of the peptide were degraded by trypsin, but there remained a protected band of >40 residues that coincides with residues shown inside the TolC channel in the cryoEM structure.

The observation in most cases of just a single punctum from fluorescently-labeled colicin peptides is puzzling, although the presence of a fluorescent signal is clearly BtuB and TolC-dependent. Frankly, I'm not sure what we learn from these experiments. Somehow the videos referred to did not come through with the copy of the manuscript I received.

The new experiments with P110A and T∆1-40 are interesting and somewhat surprising. There is nothing in the Methods about how either of those mutants was generated.

Throughout the manuscript, reference is made to the TolA box of colicins E3 and E9. However, neither of those colicins actually has a TolA box! Beginning with the early work of Bouveret et al. (Mol. Micro 1998 27, 143-157) and subsequently by others, it has been shown that what had been called the TolA box in earlier work is actually a TolB box. Colicins E2-E9 and A all interact with TolA indirectly, by binding to TolB in the periplasm, and TolB then engages TolA. The fullest picture of those interactions is in the recent work of Francis et al. (2021), which is cited in this manuscript. The only E colicin known to have a TolA box is E1. A reference (probably Pilsl and Braun, 1995) is needed when referring to the E1 TolA box. All references to a TolA box in colicins E3 or E9 need to be fixed.

I have taken the liberty of making a number of additions and corrections, in italics, to add some balance to the descriptions of the two models. Obviously, these changes are just suggestions, but I hope the authors feel they improve the paper, which is a very important addition to this field. (Note: unfortunately I see that in copying my Word document, formatting was lost so the italics cannot be distinguished from the original text.)

---

## [Author Response]

Essential revisions:1. Membrane localization experiments with ColE1-T and ColE1-TR are described in Figure 4. However, more experimental details need to be included in the figure legends and Methods, especially regarding protein concentrations in the experiments shown in Figure 4. For example, what is analysed in intact cells? Is bound toxin washed off into the supernatant? If the toxin has very high affinity will it come off the OMP receptor(s)? The relative concentrations of the colicin constructs used is not given, so there's no way to easily interpret some of the data.

We apologize for the lack of clarity for this method. We have updated the figure 4B figure caption to read:

“(B) Extracellular protease digestion assay with two colicin E1 truncation constructs incubated with *E. coli*, each construct is labeled with a C-terminal His-Tag. Periplasmic SurA was used as a membrane integrity control. After incubation with colicin fragments, cells were left intact or lysed with lysozyme. Proteins degraded by trypsin in intact cells are not protected by the outer membrane.”

The following has been added to the results:

“We assessed whether ColE1-T and ColE1-TR translocate into the cell or localize on the cell surface using an extracellular protease digestion assay (Besingi and Clark, 2015). Colicin fragment stalling on the outer membrane surface would render the colicin fragments susceptible to digestion when trypsin is in the extracellular environment. Conversely, if the colicin fragment translocates into the intracellular environment, the outer membrane will shield the protein from digestion by trypsin.

Colicin fragment localization was probed through western blot analysis of specific colicin fragments with C-terminal polyhistidine-tags. Cells were incubated with these colicin fragments and subsequently exposed to two different concentrations of trypsin. If the protein enters the cell, the outer membrane shields it from trypsin digestion, and a corresponding band is present on the western blot. Periplasmic protein SurA was used as a membrane integrity control. SurA digestion indicates that the periplasm was accessible to trypsin and a SurA band indicates that the periplasm was shielded from trypsin. As a trypsin activity control, half of the cell sample was lysed (lysed cell condition) exposing all cellular compartments to trypsin.”

And the following was added to the methods:

“Extracellular Protease Digestion.

*Growth, incubation with protein, and washing:* 5 mL overnight cultures of *E. coli* (BW25113) were grown in LB meda at 37 °C with shaking (250 r.p.m.) overnight. A 50 mL culture of LB was inoculated with 1 mL of the overnight culture and grown at 37 °C with shaking (250 r.p.m.) until an OD600 of 0.6 was reached. The cultures were centrifuged at 4,000 g at 4°C for 10 minutes. For the exogenous addition of protein, cell pellets were resuspended in cold PBS (Bio-Rad) containing 10 μM of either ColE1-T or ColE1-TR protein. Cells were incubated with protein with gentle rocking for 1 hour at 37 °C. After incubation with protein, the cells were collected at 4,000 g at 4 °C for 10 minutes and resuspended in cold PBS without protein. The centrifugation and wash were then repeated a second time. The PBS washes are designed to remove unbound protein.

*Protease digestion of intact cells and lysed cells*: Each sample is split into two-1 mL aliquots. The first aliquot is the intact cell condition which is placed on ice. The second aliquot is the lysed cells control for protease activity. This aliquot was lysed by adding 0.25 mg/mL lysozyme (10 μL of a 25 mg/mL in 10 mM Tris pH 8.0, 50% (v/v) glycerol) and incubated at room temperature for 15 minutes followed by five freeze-thaw cycles in liquid nitrogen. The intact cells condition and lysed condition were both split into three 270 μL aliquots. For each triplet, trypsin was added at 5 and 20 5 and 20 μg/mL s to digest any extracellularly exposed protein. 30 μL of PBS was added to the third aliquot of each triplet for a no trypsin sample. The three aliquot were incubated at room temperature to allow for trypsin digestion. After 30 minutes, 20 μL 100 mM PMSF was added to terminate the trypsin digestion reaction. Then the samples associated with the intact cells condition were lysed. Lysozyme was added to a final concentration of 0.25 mg/mL and incubated at room temperature for 15 minutes followed by five freeze-thaw cycles in liquid nitrogen. All samples were stored at -20 °C until ready for western-blot analysis.

*Western blot:* Samples were separated by SDS-PAG *E. coli* cin proteins were probed with THE HisTag mAb mouse (GenScript). SurA was probed with anti-SurA Rabbit polyclonal (Cusabio). Western blots were imaged by fluorescence using a LI-COR Odyssey imager with secondary antibodies IRDye 800CW goat anti-mouse IgG and IRDye 800CW Donkey anti-Rabbit for Colicins and SurA, respectively. “

Furthermore, Figure 4C shows one cell each, and one could argue that only the best data is shown involved. Could the authors (perhaps as source data) show panels with many cells to show any differences between strains more objectively?

To address this question, we have added figure 4, supplemental figure 1 which contains images that show the variety of the original 315 images that give rise to the data summarized in Figure 4C-D. 18 Phase contrast and fluorescence images of cells are shown (ColE1-TR-Cy3 with 6 WT, 6 Δ*tolC*, and 6 Δ*btuB* cells).

2. The Discussion section needs to better describe the differences between the "pillar" and "total-thread" models, perhaps by also including a schematic of both. Furthermore, the experimental evidence in this paper supports the total-thread model, whereas the more ambiguous experiments (trypsin digestion) are only suggestive of the pillar model. Please make the final proposed model clearer in the discussion and include how the downstream pore-forming cytotoxic domain is translocated.

We have updated our discussion of the pillar vs. total thread model as follows and have added the following schematic to show the difference:

“There are two conflicting models of ColE1 translocation: (1) the total thread model in which the entire colicin unfolds and threads through TolC (Cramer *et al.*, 2018, Jakes, 2017) and (2) the pillar model, in which colicin E1 inserts into TolC as a helical hairpin, facilitating LPS-mediated, self-translocation of the colicin cytotoxic domain (Zakharov *et al.*, 2016). Our data support aspects of both models. The belief in a closed hinge conformation of bound ColE1 prompted two arguments against the total thread mechanism: (1) the ColE1-T closed hinge conformation is too wide to fit through TolC and (2) both ends would face the extracellular milieu (Zakharov *et al.*, 2016, Cramer *et al.*, 2018). The unanticipated ColE1-T hinge opening in the bound state resolves these objections: colicin threads into the TolC barrel with the amino terminus pointing into the periplasm, as the total thread model hypothesized. Our trypsin digests show ColE1-TR stalls at the outer membrane of *E. coli* cells and the single-molecule fluorescence images confirm that ColE1-TR remains clustered and do not diffuse as would be expected if entering the periplasm or cytoplasm. These puncta form regardless of the presence or absence of the TolA binding region on ColE1-TR.

Though we cannot exclude the possibility that ColE1 T and R domain translocation depends on the C domain, we observe—as hypothesized by the pillar model—that colicin sits inside TolC but does not translocate. These results are in agreement with early studies of pore-forming colicins in which trypsin added to the extracellular environment can reverse colicin activity (Plate and Luria, 1972, Dankert *et al.*, 1980, Dankert *et al.*, 1982, Benedetti *et al.*, 1992, Duche, 2007). We anticipate that the stable ColE-TR:TolC complex forms regardless of the presence of the C domain.

If the C domain does translocate while the T and R domains remain bound, we anticipate that the requirement for TolA is due to its role in membrane maintenance. Binding of TolA by a different group A colicin (ColE9) causes membrane defects similar to those in a Δ*tolA* strain (Rassam et al., 2018). Therefore, in the pillar model, colicin binding of TolA may facilitate C domain insertion through the defective membrane, whereas in the total thread model binding TolA transduces the proton motive force to unfold ColE1and pull it into the periplasm (Figure 6, supplemental figure 1). Because we did not use the toxic C domain, our data cannot clarify what occurs after the T and R domain of full length colicin bind the outer membrane proteins. Although our studies find that ColE1-TR remains on the surface of the bacteria, it is certain that the C domain, when present, translocates across the membrane into the target bacteria to kill the cell. Our data does not exclude the possibility that the translocation of the T and R domains occurs along with the C domain in full length colicin, nor does it support T or R domain translocation. Thus, though our studies give a more complete view of the early stages of ColE1 binding the surface of its target, we cannot determine if insertion of the C domain occurs via the total thread model or via a modified pillar model.

3. Experiments with GFP attached to colE1-TR suggest efficient uptake in a tolC-dependent process. What would be a very informative further would be to delete the TolA box from the N terminus of the T domain of the T-R construct and determine whether uptake still occurs. If so, that would argue further against a translocation model whereby the C-terminal channel-forming domain of colicin E1 slithers down the outside of TolC, with its C terminus first, as proposed by Cramer et al. (2018). Additional control would be to mutate the hinge-proline.

At the reviewers’ request we have deleted residues 1-40, which bind to TolA, and we have quantified single-molecule fluorescence images of ColE1-TR_Δ1-40_-Cy3 on WT, ∆*tolC*, and ∆*btuB* cells. We find that, like ColE1-TR, ColE1-TR_Δ1-40_ does remains fixed on the cell in puncta.

The new data can now be found in the manuscript as Figure 4—source data 1.

And as a gallery of the pictures as Figure 4 supplemental figure 3.

We now describe this data in the manuscript as follows:

“In binding to the cell surface, ColE1-TR_Δ1-40_ binds to WT, Δ*tolC*, and Δ*btuB* in a similar manner as ColE1-TR (Figure 4 supplementary figure 3, Figure 4 supplementary table 1). Cell localization of ColE1-TR matching that of translocation deficient ColE1-TR_Δ1-40_ suggests that ColE1-TR puncta are not within the cell and likely stalled on the outer membrane receptor.”

We also mutated the hinge proline (P110A) and found that ColE1-T still binds to TolC. This data has been added as Figure 3, supplemental figure 2.

We added the following text to the manuscript:

“These contacts are strong enough that even when the proline hinge of the TolC-box is mutated (P110A) ColE1-T still binds TolC (Figure 3, supplementary figure 2).”

4. The Housden klebicin paper was published very recently describing detailed studies of the interaction of TolC with another bacteriocin, including cryo-EM structures (Housden et al., Nature Commun 12(1):4625). While the work described here has been carried out independently, some acknowledgement of this work in the discussion is warranted.

We have added the following text to the discussion:

“Shortly after this work was made available, a similar structure demonstrated that KlebC, an rRNAse bacteriocin from *Klebsiella pneumoniae*, binds TolC from *Klebsiella quasipneumoniae* with a binding mode similar to that of ColE1 binding TolC. (Housden *et al.*, 2021) KlebC is closed in the unbound state as confirmed by the high-resolution xray crystal structure which supports our proposed conformation (Figure 2E) and opens into a single-pass, kinked-helix to bind to TolC. This suggests a conserved mechanism for bacteriocins that hijack these trimeric efflux pump proteins.”

[Editors' note: further revisions were suggested prior to acceptance, as described below.]

The manuscript has been improved but there are some remaining issues that need to be addressed, as outlined below:1. While we realise the PDBs were released earlier than those of Housden et al. it would be better to remove the instances of "first" in the text as, after all, they are not really needed.

Though we agree it is clear that our PDB was deposited and released before that of Housden et al., our concern is that the wording “first” is unfortunately needed. Housden et al. claims specifically that, “We next set out to delineate the TolC-binding region of KlebC and determine its structure; no structure has yet been reported for any TolC-binding domain.”

We are concerned Housden’s wording will perpetuate a misunderstanding in the literature as our structure had already been reported, in a preprint and a PDB. Such a misunderstanding might reduce the number of citations attributed to this work in this journal as other scientists would erroneously believe that priority should be attributed to Housden.

2. The modified text for points 23 and 24 in the previous submission was still unclear to Referee 2. We would ask you to try to re-address these points (re-pasted below for clarity).23. line 253: "need to be in close proximity for colE1 binding to occur" needs a reference.24. line 254: "when BtuB clusters together in groups of 12 or more, it may exclude TolC" needs a reference. Also, please explain "These cluster geometries would therefore lower the number of full binding sites available for the T and R domains of colicin E1."

We have rewritten this passage to make it clear that we are referring to the simple need for physical proximity during binding: if a substrate is binding two proteins simultaneously, then those two proteins must therefore be near to each other. Moreover, clustering multiple proteins in a small two-dimensional area leads to physical (geometric) exclusions:

“Because we usually only see one punctum per cell, we anticipate that some BtuB and TolC may remain unbound because of the geometric constraints of punctum formation. ColE1 must engage both TolC and BtuB simultaneously. Therefore, both receptors must be in close proximity, similar to the ~50 Å proximity of BtuB to OmpF when both bound to a another type A colicin, ColE9 (Francis et al., 2021). Given the two-dimensional surface of a membrane there are restrictions to how many proteins can be within 50 Å of any other protein. For the clusters we see to occur, when BtuB clusters together in groups of approximately 20 (Figure 4) TolC must also cluster in groups of approximately 20, ultimately requiring approximately 40 proteins in a relatively small area. The competition between the clustering of BtuB and the requirement for proximity of BtuB and TolC for binding could lower the number of BtuB/TolC sites available for the T and R domains of colicin E1 due to the relative TolC and BtuB geometries needed for ColE1 binding.”

3. Throughout the manuscript, reference is made to the TolA box of colicins E3 and E9. However, neither of those colicins actually has a TolA box. Beginning with the early work of Bouveret et al. (Mol. Micro 1998 27, 143-157) and subsequently by others, it has been shown that what had been called the TolA box in earlier work is actually a TolB box. Colicins E2-E9 and A all interact with TolA indirectly, by binding to TolB in the periplasm, and TolB then engages TolA. The fullest picture of those interactions is in the recent work of Francis et al. (2021), which is cited in this manuscript. The only E colicin known to have a TolA box is E1. A reference (probably Pilsl and Braun, 1995) is needed when referring to the E1 TolA box. All references to a TolA box in colicins E3 or E9 need to be fixed.

We thank the reviewer for this insight and have made the distinction between colicin E1 binding to TolA and colicins E3 and E9 binding to TolB. References to TolA box for colicins E3 and E9 have been corrected through the manuscript with the appropriate citations. In addition, all references to ColE1 requiring TolA now include the citation to Pilsl and Braun 1995.

The text has been updated:

“TolA is known to be required for ColE1 cytotoxicity (Nagel de Zwaig and Luria, 1967, Pilsl and Braun, 1995). ColE1 lacking the TolA box at the N-terminal T-domain would likely be translocation incompetent due to uncoupling from TolA. Similarly, there is a loss of biological activity due to mutations in the ColE9 N-terminal region that indirectly interacts with TolA (Loftus et al., 2006).(Schneider et al., 1997) Biological inactivation has also been caused by mutation to the Tol-Pal interacting regions of ColE3 (Escuyer and Mock, 1987) and ColA (Pilsl and Braun, 1995).”

“TolA is normally involved in outer membrane invagination during cell division (Gerding et al., 2007). Like Colicin E1, Colicin E9 contains an intrinsically unstructured N-terminus that engages the Tol-Pal system once it penetrates into the periplasm (Loftus et al., 2006). Engagement of the Tol-Pal system by the N-terminal region of colicin E9 disrupts native Tol-Pal system function leading to outer membrane defects (Rassam et al., 2018).”

“Recruitment of the Tol-Pal system by a different group A colicin (ColE9) causes membrane defects similar to those in a ΔtolA strain (Rassam et al., 2018).”

4. While the experimental details and source data have been expanded and clarified, the implications of both the trypsin digestion experiments and fluorescence labeling are less clear than the cryo-EM and drug efflux experiments, and some caveats should be added to the discussions of those experiments.

We thank the reviewer for the helpful discussions of recent findings. We agree that the single puncta were a surprising finding and worth pointing out in our manuscript. Though the molecular mechanism that gives rise to these puncta remains to be explored in future projects, the error bars and detection limits of the microscopy observations indicate that this clustering is true.

We also appreciate that the results of our in vivo trypsin digest were (and were expected to be) different from the recent Housden work in which trypsin digests were conducted in vitro absent any mechanism for translocation. Finally, we appreciate the reviewer for pointing out that we neglected to update our methods on the P110A mutant and the del1-40 mutant.

We have added the following to the methods:

“A gBlock (Integrated DNA Technologies) was synthesized for GFP containing complimentary flanking sequences to the plasmid with ColE1-TR and inserted with the omega-PCR technique.

ColE1-TRΔ1-40 was produced by inverse PCR to delete residues corresponding to 1 through 40.

All point mutants (ColE1-T-E192C, ColE1-TR-E366C, ColE1-TR-E366CΔ1-40, and ColE1-T-P110A) were generated by inverse PCR with mutagenic primers.”

5. The argument between the two competing models may require some further clarification. Referee 3 has made some suggestions below. Please look at these and amend appropriately as you see fit.Specific comments and corrections:Line 62: …targets and killing mechanisms. Most colicins…

Done.

Lines 69-71 should read: …of ColE3 or ColE9 initiates translocation using the secondary OMP receptor/translocator, OmpF, to access the Tol-Pal system in the periplasm. For most colicins, the T domain requires an OMP distinct from the R domain target for colicin translocation.

Done.

Line 72-73: …structurally characterized bound to their OMP primary receptors. (Kurisu…, Sharma…, Buchanan et al., 2010). Those structures all reveal that the plugged β-barrels remain plugged upon colicin binding, so another route through the outer membrane is required.

We appreciate this suggestion and we have added the Buchanan reference.

Line 75-76 ff: …that T domain can pass fully through one pore of its membrane translocator, OmpF, and loops around to insertv in a second pore of the trimer from the periplasmic side, while the R domain initially interacts with…its receptor, BtuB. Those same studies and earlier ones clearly demonstrated that full engagement of the E9 T domain with its TolB translocation partner in the periplasm results in disengagement of the R domain from BtuB (Francis et al. 2021 and references therein). Experiments such as these and those that preceded this most recent work are the basis for the "total thread" model for colicin translocation, in which the colicin is pulled through the same pore of OmpF through which its T domain initially entered. The process is energized by the TolABQR system.

We appreciate this level of detail for the insertion mechanisms of all colicins but think that such descriptions might be better suited for future review articles.

Line 82: death. Unlike any of the other E colicins, the T domain of colicin E1…

Done.

Line 83: …and the R domain binds BtuB, the primary receptor shared by all E colicins.

Done.

Line 98: …and disrupt TolC channel conductance in vitro, in planar lipid bilayer membranes…

Done.

Line 101: …in solution similar to another colicin T domain from colicin Ia, one that does not interact with TolC (Wiener et al., 1997).

Done.

Line 223: …these residues are contact sites…

Done.

Line 290: Should read: Given the two-dimensional surface…

Done.

Line 370: Add Francis et al., 2021 reference here.

Done.

Line 376: Why was the hinge opening unanticipated?

We have updated the text to explain more clearly:

“The ColE1-T hinge opening in the bound state—which was unanticipated by the pillar model— resolves these objections”

Line 378: …as the total thread model hypothesized and consistent with experimental data that support this model for colicin E9 (Housden et al., 2005; Housden et al., 2013, Francis et al., 2021).

Because of the following discussion in the manuscript:

“the “total thread” model (Cramer et al., 2018, Zakharov et al., 2016, Housden et al., 2013, Francis et al., 2021) posits that the protein unfolds and passes through TolC N-terminus-first as an unstructured peptide and binds to TolA (Pilsl and Braun, 1995) in the periplasm. In this model the binding between the intrinsically unstructured colicin N-termini and periplasmic proteins (Jakes, 2017, Housden et al., 2013) creates a pulling force that results in the translocation of the whole colicin.”

We have not added the requested text out of concern that adding it would be overly repetitive and may frustrate the reader.

Line 380: …clustered and does not diffuse…

Done.